# Exploration of Oxidative Chemistry and Secondary Organic Aerosol Formation in the Amazon during the Wet Season: Explicit Modeling of the Manaus Urban Plume with GECKO-A

Camille Mouchel-Vallon[1,a], Julia Lee-Taylor[1,2], Alma Hodzic[1], Paulo Artaxo[3], Bernard Aumont[4], Marie Camredon[4], David Gurarie[5], Jose-Luis Jimenez[2,6], Donald H. Lenschow[7], Scot T. Martin[8,9], Janaina Nascimento[10,11], John J. Orlando[1], Brett B. Palm[2,6,b], John E. Shilling[12], Manish Shrivastava[12], and Sasha Madronich[1]

[1]Atmospheric Chemistry Observations and Modeling, National Center for Atmospheric Research, Boulder, CO 80301, USA
[2]Cooperative Institute for Research in Environmental Sciences (CIRES), University of Colorado, Boulder, CO 80309, USA
[3]University of Sao Paulo, Institute of Physics, Rua do Matao 1371, 05508-090, Sao Paulo, S.P., Brazil
[4]LISA, UMR CNRS 7583, Université Paris-Est-Créteil, Université de Paris, Institut Pierre Simon Laplace, Créteil, France
[5]Department of Mathematics and Center for Global Health and Diseases, Case Western Reserve University, Cleveland, OH 44106-7080, USA
[6]Department of Chemistry, University of Colorado, Boulder, CO 80309, USA
[7]Mesoscale and Microscale Meteorology Laboratory, National Center for Atmospheric Research, Boulder, CO 80301, USA
[8]School of Engineering and Applied Sciences, Harvard University, Cambridge, MA 02318, USA
[9]Department of Earth and Planetary Sciences, Harvard University, Cambridge, MA 02318, USA
[10]Post-graduate Program in Climate and Environment, National Institute for Amazonian Research and Amazonas State University, Manaus, AM, Brazil
[11]Chemical Sciences Division, NOAA Earth System Research Laboratory, Boulder, CO 80305, USA
[12]Pacific Northwest National Laboratory, Richland, WA 99352, USA
[a]Now at Laboratoire d'Aérologie, Université de Toulouse, CNRS, UPS, Toulouse, France
[b]Now at Department of Atmospheric Sciences, University of Washington, Seattle, WA 91895, USA

**Correspondence:** C. Mouchel-Vallon (camille.mouchel-vallon@aero.obs-mip.fr)

**Abstract.** The GoAmazon 2014/5 field campaign took place in Manaus (Brazil) and allowed the investigation of the interaction between background level biogenic air masses and anthropogenic plumes. We present in this work a box model built to simulate the impact of urban chemistry on biogenic Secondary Organic Aerosol (SOA) formation and composition. An organic chemistry mechanism is generated with the Generator for Chemistry and Kinetics of Organics in the Atmosphere (GECKO-A) to simulate the explicit oxidation of biogenic and anthropogenic compounds. A parameterization is also included to account for the reactive uptake of isoprene oxidation products on aqueous particles. The biogenic emissions estimated from existing emission inventories had to be reduced to match measurements. The model is able to reproduce ozone and $NO_x$ for clean and polluted situations. The explicit model is able to reproduce background case SOA mass concentrations but is not capturing the enhancement observed in the urban plume. Oxidation of biogenic compounds is the major contributor to SOA mass. A Volatility Basis Set parameterization (VBS) applied to the same cases obtains better results than GECKO-A for predicting SOA mass in the box model. The explicit mechanism may be missing SOA formation processes related to the oxidation of monoterpenes that could be implicitly accounted for in the VBS parameterization.

# 1 Introduction

The Amazonian rainforest is the largest emitter of biogenic primary hydrocarbons on Earth (*e.g.* Guenther et al., 2012). Pho-
tochemistry in this tropical region is more photochemically active than other regions throughout most of the year, which
stimulates the oxidation of the biogenic primary compounds by oxidants such as ozone and OH radicals. This part of the world
is consequently a substantial source of Secondary Organic Aerosol (SOA) (Martin et al., 2010; Chen et al., 2015a) produced
by condensation of oxygenated secondary organic species formed from gas and aqueous phase oxidation of biogenic com-
pounds (Claeys, 2004; Carlton et al., 2009; Paulot et al., 2009). On the other hand, the city of Manaus (Brazil) is a source
of anthropogenic pollution with 2.1 million inhabitants, ca. 600000 vehicles in circulation and 78 thermal power plants in its
close surroundings (Abou Rafee et al., 2017). Manaus is situated at the confluence of the Rio Negro and Solimões rivers that
subsequently form the Amazon River (Fig. 1). This metropolis is isolated from the rest of South American populated areas by
over 1000 km of Amazonian tropical rainforest in every direction (*e.g.* Martin et al., 2016). Manaus is therefore a point source
of urban pollution in a vast rainforest, making it an ideal place to study chemical interactions of biogenic and anthropogenic
compounds. The Observations and Modeling of the Green Ocean Amazon (GoAmazon 2014/5) experiment was designed to
characterize the anthropogenic perturbations to the clean air masses influenced by Amazonian natural emissions (Martin et al.,
2016). The main instrumented site (T3) was situated approx. 70 km southwest of Manaus (see Fig. 1). In addition, the US De-
partment of Energy (DOE) Gulfstream research aircraft (G-1) conducted 16 research flights to sample the Manaus plume as it
was transported downwind and over the Amazon forest (Martin et al., 2016; Shilling et al., 2018). With varying meteorological
conditions, this allowed sampling of clean background air from the Amazon basin and polluted air from Manaus (Martin et al.,
2016).

Several studies have already shown that the overall composition of particulate matter (PM) in remote areas is distinctly
different from urban areas, with anthropogenic PM being characterized by more sulfates and hydrocarbon-like compounds,
whereas remote PM contains more oxidized organic matter (e.g. Xu et al., 2015; Budisulistiorini et al., 2016). In the Manaus
environment, biogenic molecules would interact with the chemistry resulting from anthropogenic emissions. de Sá et al. (2018)
have shown that the majority of sub micrometer particle mass at the T3 site is secondary. Several studies have investigated
how the biogenic nature of the SOA is affected by anthropogenic influence. For instance Aerosol Mass Spectrometer (AMS)
measurements reported by de Sá et al. (2017) have shown that the contribution of epoxydiols derived from isoprene to SOA
(IEPOX-SOA) amounts to 11 to 17% of the total organic mass when the Manaus plume is sampled, compared to 19 to 26%
under background conditions. Using an Oxidation Flow reactor (OFR) and tracers for different source types, Palm et al. (2018)
concluded that the Volatile Organic Compounds (VOC) and Intermediate Volatility Organic Compounds (IVOC) sampled
during GoAmazon2014/5 could form SOA whose origin would be dominated by biogenic sources during the dry season, and
by both biogenic and anthropogenic sources during the wet season. With a regional model study of the GoAmazon 2014/5
situation, Shrivastava et al. (2019) concluded that the higher oxidative capacity in the urban plume results in an enhancement
of biogenic SOA production.

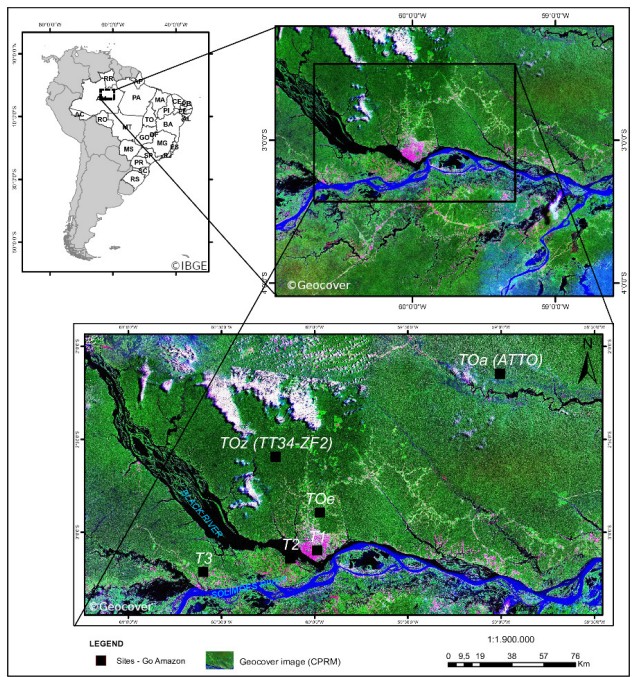

**Figure 1.** Map of the GoAmazon field campaign instrumented sites. Measurements used in this work came from the T3 site. ©Geocover, ©IBGE.

Models need to take into account the different nature of VOCs and SOA resulting from biogenic and anthropogenic chemistry to accurately represent their interactions. This can be done by looking at this problem with what Pankow et al. (2015) call a "molecular view", as opposed to the "anonymized view" followed by current 3D models. The molecular view attempts to predict SOA mass from the known and estimated properties of individually simulated organic compounds while the anonymized view uses hypothetical properties (*e.g.* volatility, solubility) of a small number of lumped compounds. In a recent review, Heald and Kroll (2020) have reported on the recent progress in measurements of individual organic compounds, and how experimentalists are getting close to achieving closure on organic carbon in both gas and aerosol phases (*e.g.* Gentner et al., 2012; Isaacman-Vanwertz et al., 2018). As these measurements are now able to capture elemental formulas, double bonds, some oxygenated functional groups and aromaticity (*e.g.* Yuan et al., 2017), they still do not provide individual molecular identities. From this point of view, measurements are still restricted to a "formula view". For the GoAmazon field campaign, Yee et al. (2018) were able to sample and identify 30 sesquiterpenes and 40 of their oxidation products at the T3 site with a semi-volatile thermal desorption aerosol gas chromatograph (SV-TAG, Isaacman et al., 2014) but they do not achieve the coverage needed to approach the "molecular view".

3D models that were run for the Manaus situation offer an anonymized view of SOA composition (Shrivastava et al., 2019) because they rely on a Volatility Basis Set parameterization (VBS, Donahue et al., 2006). The Generator for Explicit Chemistry and Kinetics of Organics in the Atmosphere (GECKO-A, Aumont et al., 2005; Camredon et al., 2007) is an excellent tool

to model atmospheric organic chemistry with a detailed molecular view. GECKO-A is an automated chemical mechanism generator built to write the explicit chemistry of given precursors by following a prescribed set of systematic rules. This set of systematic rules relies on experimental data when available and Structure Activity Relationships (SAR) to determine unknown kinetic or thermodynamic constants. It has previously been run in box models to evaluate processes like secondary organic aerosol formation (Valorso et al., 2011; Aumont et al., 2012; Camredon and Aumont, 2006; Camredon et al., 2007) and dissolution of organic compounds (Mouchel-Vallon et al., 2013). It was also applied to simulate chamber experiments (Valorso et al., 2011; La et al., 2016) and urban and biogenic plumes (Lee-Taylor et al., 2011, 2015).

In this work, a box model is run to simulate the evolution of an Amazonian air mass intercepting Manaus emissions during the wet season. Emissions of anthropogenic and biogenic primary VOCs are estimated with available data. The chemical scheme describing the explicit oxidation of these primary compounds is generated with GECKO-A. The resulting detailed simulation is then used to explore the impact of Manaus emissions on the Amazonian biogenic chemistry. Comparisons with aerosol mass spectrometer data and the VBS parameterization are carried out to identify important processes involved in biogenic SOA formation that may not be accounted for in GECKO-A. Finally the potential for reduction of the explicit mechanism is estimated.

## 2 Experimental Data

The main instrumented site (referred to as T3 hereafter) of the GoAmazon 2014/5 field campaign was situated 70 km west of Manaus (Fig. 1). Two aircraft were also deployed, a G-159 Gulfstream I (G-I) (Schmid et al., 2014) that flew at low altitude and mostly sampled the boundary layer and a Gulfstream G550 (HALO) that flew higher altitudes and sampled the free troposphere (Wendisch et al., 2016). The flight tracks are depicted in Martin et al. (2016) and Wendisch et al. (2016). The G-1 airplane mainly flew daytime transects of the Manaus plume between the city and the T3 site.

The detailed instrumentation deployed at T3 and in the airplanes has been described elsewhere (Martin et al., 2016). For this study we mainly relied on ground deployed instruments briefly described here.

Ozone concentration measurements made with a Thermo Fisher Model 49i Ozone Analyzer were obtained from the Mobile Aerosol Observing System-Chemistry (MAOS-C).

Due to some issues with the $NO_x$ analyzer deployed at T3 by the MAOS-C during the wet season, $NO_x$ data reported here is weakly reliable. The values reported here are only qualitative indications of $NO_x$ levels in the studied period.

OH radicals concentrations were provided by an OH chemical ionization mass spectrometer (Sinha et al., 2008, OH-CIMS).

Organic compounds in the gas phase were measured with selected reagent ion proton transfer reaction time-of-flight mass spectrometer (SRI-PTR-ToFMS, Jordan et al., 2009a, b). Aerosol composition was monitored by a high-resolution time-of-flight aerosol mass spectrometer (HR-ToF-AMS) (DeCarlo et al., 2006; de Sá et al., 2018, 2019).

For the purpose of comparisons with the model, we need to be able to separate time periods representing clean and polluted episodes. Using a fuzzy c-means clustering algorithm (Bezdek, 1981; Bezdek et al., 1984) applied to T3 measurements, de Sá et al. (2018) were able to identify four different clusters corresponding to (i) fresh or (ii) aged (2+ days) biogenic production,

**Table 1.** Box model constraints used in the clean and polluted setups

| | Clean Background | Manaus |
|---|---|---|
| NO soil emission [molec cm$^{-2}$ s$^{-1}$][a] | $8.3 \times 10^9$ | – |
| Aerosol number concentration [cm$^{-3}$][b] | $5 \times 10^2$ | $1 \times 10^4$ |
| Aerosol pH | 3.0 | 1.5 |
| Aerosol sulfate concentration [μg m$^{-3}$][b] | 0.3 | 0.4 |
| Aerosol nitrate concentration [μg m$^{-3}$][b] | 0.05 | 0.1 |
| Hygroscopicity Parameter ($\kappa$)[c] | 0.15 | 0.15 |

[a]Shrivastava et al. (2019) [b]de Sá et al. (2018) [c]Thalman et al. (2017)

and air masses influenced by the (iii) northern or (iv) southern parts of Manaus. Using the timeseries contribution of these clusters, we labeled as background air masses that were identified as being composed of at least 50% of any clean cluster (i or ii). Conversely, air masses that were identified by de Sá et al. (2018) as being composed of at least 50% of any polluted cluster (iii and iv) were labeled as polluted. The clustering methods constrained the classification to only include wet season afternoon air masses that were not exposed to rain in the previous day (see de Sá et al., 2018). These limitations match with 100 our model restrictions which do not include cloud chemistry, nor fire emissions that would be important during the dry season. For comparison with the model, experimental data were hourly averaged for each cluster.

## 3 Model Setup

A Lagrangian box model was built to simulate chemistry in the planetary boundary layer and the residual layer for an air parcel traveling over the Amazonian forest and Manaus. Because experimental data compared to the model only contain air 105 masses that were not exposed to rain in the previous day (see Sect. 2 and de Sá et al., 2018), the model simulates biogenic conditions for one day, assuming the air mass was washed out by rain prior to that day. After the one day spinup, biogenic emissions are replaced by urban emissions for one hour during the second day to represent the interaction of the air mass with the Manaus urban area. After the simulated encounter with Manaus, the model inputs return to biogenic emissions until the end of the second day. This simulation is defined hereafter as the "polluted" case. Another simulation is run where the box is 110 only subjected to biogenic emissions for two days, without any exposure to urban emissions to simulate a background case. This simulation is defined hereafter as the "clean" case. This section describes the box model setup, how the emissions were defined and the chemical mechanism used for this study.

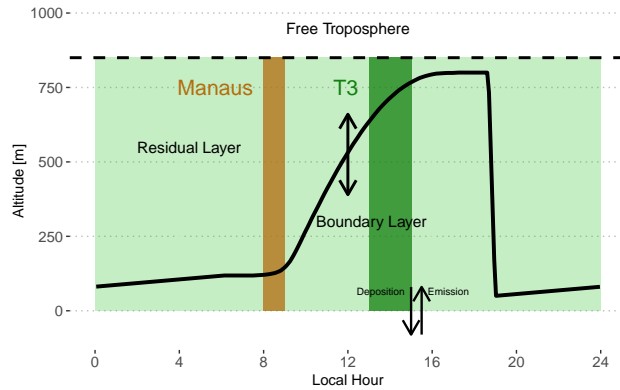

**Figure 2.** Schematic depiction of the box model setup used in this work. The continuous black line shows the time evolution of the PBL height. The dashed black line depicts the top of the residual layer box. The brown shaded area is the period when the box is subjected to Manaus emissions. For the rest of the time period, the box is subjected to biogenic emissions (light and dark green shaded areas). The dark green shaded area is approximately the period when the box would be over the main instrumented site T3, assuming a travel time of 4 to 6 hours.

## 3.1 Box model

This study relies on a box model described in this section. It includes emissions from the forest and the city, deposition and
115 chemical evolution of the trace gases. Daytime growth of the planetary boundary layer is also simulated, with mixing with the residual layer.

### 3.1.1 Boundary Layer

The model includes two boxes on top of each other separated by a moving boundary representing the height of the boundary layer. The bottom box extends from the surface to the top of the planetary boundary layer (PBL). The top box extends from
120 the top of the planetary boundary layer to 850 m and represents the residual layer (RL) (see Fig. 2). The daytime PBL height evolution is parameterized according to the Tennekes (1973) approach and was calculated using the Second-Order Model for Conserved and Reactive Unsteady Scalars (SOMCRUS, Lenschow et al., 2016) (see Fig. 2). At sunset, stratification is assumed to quickly shrink the PBL to 50 m which results in the contents of the PBL being reallocated to the RL. During the night, the PBL is constrained to linearly grow to reach the next morning level. The PBL height evolution is the same for each of the two
simulated days. During the day, the PBL is therefore slowly incorporating residual chemicals resulting from the previous day and night chemistry. Thalman et al. (2017) report PBL heights estimated from ceilometer measurements during the wet season in the central Amazonian Forest, for polluted and background conditions. The measurements reach a maximum of 800 m at around 3pm local time. This value was used to further constrain the PBL height evolution by scaling the SOMCRUS output to reach this measured PBL height maximum. The growth and shrinking of the PBL dilute the expanding box and transfer gases
from the shrinking box to the expanding box. This is parameterized according to Eqs. 1 and 2:

$$\frac{dC_b}{dt} = \begin{cases} \frac{1}{h}\frac{dh}{dt}C_t - \frac{1}{h}\frac{dh}{dt}C_b & \text{if } \frac{dh}{dt} > 0 \\ 0 & \text{if } \frac{dh}{dt} \leq 0 \end{cases} \tag{1}$$

$$\frac{dC_t}{dt} = \begin{cases} 0 & \text{if } \frac{dh}{dt} \geq 0 \\ -\frac{1}{H-h}\frac{dh}{dt}C_b + \frac{1}{H-h}\frac{dh}{dt}C_t & \text{if } \frac{dh}{dt} < 0 \end{cases} \tag{2}$$

$C_b$ and $C_t$ [molec cm$^{-3}$] are chemical species concentrations in the PBL (bottom) and RL (top) boxes respectively. $h$ [m] is the variable height of the PBL and $H$ [m] is the fixed altitude of the RL top. The first term in each equation describes the addition of material coming from the shrinking box and the second term describes the dilution of the growing box. Following these equations, mixing happens in two stages: (i) the long RL entrainment into the PBL over day time and (ii) the rapid transfer of the PBL to the RL at sunset. The box model approach assumes rapid mixing in both layers and that chemistry is applied to well-mixed concentrations. The residual layer is also slowly mixed with the free troposphere. The free troposphere is assumed to be a fixed reservoir of CO (80 ppb) and ozone (15 ppb, *e.g.* Browell et al., 1990; Gregory et al., 1990; Kirchhoff et al., 1990). The subsidence velocity is constant and fixed at 0.1 cm s$^{-1}$ (*e.g.* Raes, 1995).

Temperature is assumed to follow a sinusoidal daily variation, with an average of 27 °C, an amplitude of 4 °C and a maximum at 6 pm local time. Relative humidity is initially set at 75% at 6 am (23 °C) and is free to evolve with temperature changes assuming water vapor concentration is constant.

## 3.2 Emissions

### 3.2.1 Biogenic Emissions

VOC emissions from the rainforest were estimated with the Model of Emissions of Gases and Aerosols from Nature (MEGAN v2.1, Guenther et al., 2012). Biogenic emissions on March, 13$^{\text{th}}$ 2014 (the golden day of the GoAmazon field campaign, see de Sá et al., 2017) in a domain situated in the forest around Manaus were averaged to obtain total isoprene and monoterpene hourly averaged emissions for a day typical of the wet season, without any recorded rain event. Monoterpenes were then speciated to match concentrations measured by Jardine et al. (2015) at the top of an Amazonian rainforest canopy with a thermal desorption-gas chromatograph-mass spectrometer (TD-GC-MS). Based on this emission inventory, we then simultaneously optimized isoprene and total monoterpenes emissions to match the model with isoprene and total monoterpenes measured at T3 under clean conditions. This resulted in the need to reduce isoprene emissions by a factor of 7. Using measurements from a similar site in Amazonia, Alves et al. (2016) reported that MEGAN 2.1 overestimated isoprene emissions by a factor of 5 on average during the dry season. They assumed that the T3 site configuration (a clearing in the forest, near a road) could affect local isoprene concentrations compared to average Amazonian emissions. For instance measurements in the Amazon rainforest by Batista et al. (2019) indicate that biogenic emissions exhibit large intermediate scale heterogeneity, with estimated emission variations of 220% to 330%. Recent satellite based estimates of biogenic emissions also reported that MEGAN overestimates isoprene emissions in Amazonia by 40% (Worden et al., 2019). In a similar way, monoterpenes emissions had to be reduced by

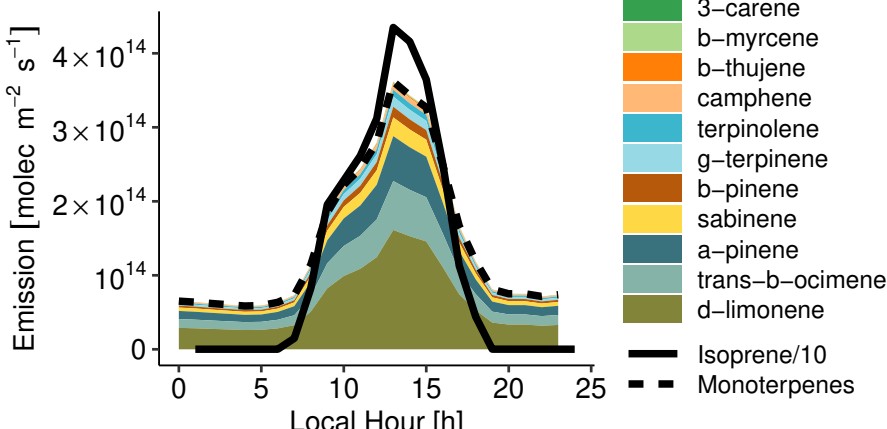

**Figure 3.** Hourly biogenic emissions estimated with MEGAN and scaled to match measured concentrations (see 3.2.1). The lines depict isoprene (continuous line) and total monoterpenes (dashed line). The colored areas depict the contribution of each individual species to total monoterpenes. Please note that isoprene emissions are divided by 10 to fit on the plot.

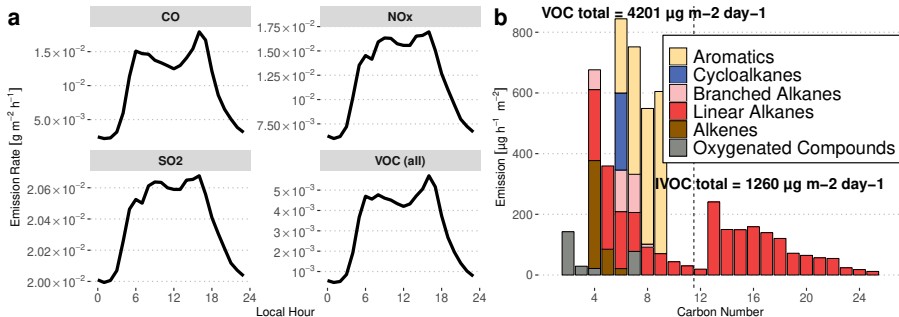

**Figure 4.** Diurnal evolution of simulated traffic emissions in Manaus deduced from inventories in Manaus and Saõ Paulo. (a) $NO_x$, $SO_2$, CO and total VOC daily emissions. (b) Carbon number distribution of Manaus emissions at noon. Total daily emissions are indicated for lighter organic compounds (VOC) and less volatile compounds (IVOC). The dashed line denotes the separation between VOCs (left) and IVOCs (right).

a factor of 8 compared to the MEGAN values. Figure 3 depicts the resulting daily biogenic emission cycle. Isoprene emissions dominate monoterpene emissions by approximately an order of magnitude. $\delta$-limonene is the most emitted monoterpene (45%), followed by trans-$\beta$-ocimene (18%) and $\alpha$-pinene (17%). NO soil emissions are also accounted for with a constant flux of $8.3 \times 10^9$ molec cm$^{-2}$ s$^{-1}$ following Shrivastava et al. (2019).

### 3.2.2 Manaus Emissions

The emissions used to represent the influence of Manaus are shown in Fig. 4a and were calculated following the methodology described in Abou Rafee et al. (2017) and Medeiros et al. (2017). Traffic emissions have been estimated from vehicle use intensity and emission factors for CO, $NO_x$, $SO_2$ and VOCs depending on type of fuel use in Manaus (Abou Rafee et al., 2017). VOC speciation is assumed to be similar to the average speciation of the vehicle fleet emissions of São Paulo, Brazil in 2004 (Martins et al., 2006). Hourly distribution of the traffic emissions is considered to be similar to the hourly traffic

distribution in São Paulo (Andrade et al., 2015). In the past decades, Brazil has become known for pioneering the large scale use of ethanol based biofuels. However, due to its isolation and being distant from south Brazilian biofuel producing regions, Manaus traffic doesn't involve consumption of significant amounts of ethanol-based fuel.

The difference in the fuel blend used in São Paulo and Manaus can introduce errors in the traffic emissions VOC speciation. For instance, a recent study by Yang et al. (2019) showed that the combustion of fuels with higher ethanol content emits

significantly less carbon monoxide and more acetaldehyde. Schifter et al. (2020) showed similar results, and also suggested that ethanol blends emit smaller amounts of simple aromatic compounds (*e.g.* benzene, toluene). This speciation uncertainty can especially have an impact on oxidants concentrations. Schifter et al. (2020) reported for instance that fuels containing ethanol would potentially produce less ozone after the oxidation of emitted organic species than fuels without ethanol. Moreover, the lifetime of OH is likely to change depending on the speciation of emitted VOCs due to varying reactivities with respect to OH.

In the same way that the potential for ozone formation could depend on the use of ethanol fuel blends, it is also possible that the potential for SOA formation would depend on these fuel blends too.

This traffic emission estimate does not include Intermediate Volatile Organic Compounds (IVOC) which would mainly be produced by diesel vehicle emissions (Gentner et al., 2012, 2017). Zhao et al. (2015, 2016) showed that the IVOC to VOC emissions ratio lies between 4% for gasoline vehicles and 65% for diesel vehicles. Knowing that diesel vehicles account for

ca. 45% of the total driven distance in Manaus (Abou Rafee et al., 2017), we therefore assume that IVOC total emissions are approximately equal to 30% of total VOC emissions. To estimate the distribution of species resulting from IVOC emissions, we assumed that the distribution in volatility is similar to the distribution used to simulate traffic emissions in Mexico City in Lee-Taylor et al. (2011), with n-alkanes from $C_{12}$ to $C_{25}$ acting as surrogates for these heavier emitted organic compounds.

The resulting distribution of urban organic emissions at noon as a function of the number of carbon atoms is presented in

Fig. 4b. As reported in the Gentner et al. (2017) review, gasoline emissions have a maximum for $C_8$ species, with no emission of importance above $C_{12}$, whereas diesel vehicles emit species from $C_{10}$ to $C_{25}$, with a peak at $C_{12}$. These features are present in the emissions estimated in this work, with the gasoline peak around $C_{6-7}$ and the diesel maximum at $C_{13}$. Gentner et al. (2017) also report that half of the gasoline VOC emissions are composed of linear and branched alkanes, the other half consisting of aromatics and cycloalkanes. In our estimates of gasoline emissions ($C_{<12}$) the proportion of branched alkanes is smaller, alkenes

constitute a more important fraction of emitted $C_{4-6}$ species, branched cycloalkanes are missing, and aromatics constitute the majority of emissions of $C_{7-10}$ compounds. These differences could represent differing sources of fuels or different distributions of vehicle brands and ages. In the case of diesel emissions, Gentner et al. (2017) report that they are approximately equally

distributed between aromatics, branched cycloalkanes, bicycloalkanes and branched alkanes whereas our method leads to diesel emissions being only constituted of n-alkanes, which are used here as surrogate species for the entire mixture.

Choosing alkanes as surrogates for emitted IVOCs is likely to introduce uncertainties to SOA produced from their oxidation. Lim and Ziemann (2009) carried out multiple chamber experiments that investigated the impact of branching and rings on alkanes SOA yields. For instance they showed that SOA yields range from a few percent for branched alkanes with 12 carbon atoms to $80\%$ for cyclododecane while n-dodecane has an SOA yield of $\approx 32\%$. La et al. (2016) simulated these experiments with GECKO-A and they were able to reproduce this experimentally observed behavior. This means that without a detailed inventory of emitted IVOCs, the uncertainty on the SOA yield from IVOCs is high in our version of the model. It should be noted that the range of measured SOA yields for structurally different compounds with the same number of carbon atoms seems to peak for $C_{10}$ to $C_{13}$ alkanes. The range of observed SOA yields in Lim and Ziemann (2009) decreases after this peak. For instance, SOA yields for $C_{15}$ alkanes of various structures range from $45\%$ to $90\%$. We can therefore expect the IVOCs SOA yield to be highly sensitive to the speciation of compounds ranging from $C_{12}$ to $C_{14}$, but this sensitivity should decrease for heavier molecular weight species.

Additionally, emissions from 11 local thermal power plants (TPP) and one oil refinery located in the vicinity of Manaus were obtained from the data presented in Medeiros et al. (2017). Based on monthly statistics of fuel use in each of the TPP and the oil refinery, combined with emission factors of CO and $NO_x$ for each type of fuel (diesel, fuel oil, natural gas), we calculated CO and $NO_x$ emissions for February, March and April 2014. These total emissions were then averaged over the whole surface area of Manaus (377 km$^2$, Abou Rafee et al., 2017). Total SO2 emissions were taken from Abou Rafee et al. (2017) and added to the urban emissions for the considered Manaus area.

### 3.3    Chemical Mechanism

#### 3.3.1    GECKO-A

All emitted organic compounds were used as inputs to GECKO-A to automatically generate the chemical scheme used in this study. The GECKO-A protocol has been described in detail in Aumont et al. (2005) and updated in Camredon et al. (2007), Valorso et al. (2011), Aumont et al. (2013), and La et al. (2016). Partitioning of low volatility compounds to the aerosol phase is described dynamically as in La et al. (2016). Vapor pressures are estimated with the Nannoolal et al. (2008) structure activity relationship. As isoprene first oxidations steps have been widely studied in the literature, there is no need to automatically generate them with GECKO-A. Isoprene chemistry first two generations of oxidation were therefore taken from the Master Chemical Mechanism 3.3.1 (Jenkin et al., 1997; Saunders et al., 2003; Jenkin et al., 2015, MCM, *e.g.*). With 12 biogenic and 53 anthropogenic precursors ranging from $C_2$ to $C_{25}$, some reductions are carried out to reduce the size of the generated mechanisms. Species with an estimated vapor pressure below $10^{-13}$ atm are assumed to entirely partition to the aerosol phase so quickly that a description of their gas phase oxidation is not needed (Valorso et al., 2011). Furthermore, lower yield, longer chain species are lumped with chemically similar compounds according to a a hierarchical decision tree based on molecular structure (Valorso et al., 2011). The resulting chemical scheme contains 23 million reactions involving 4.4 million species of

which 780000 can partition into the aerosol phase. The time integration in the two-box model setup takes approximately 0.5 computing hour per simulated hour on 16 cores (Computational and Information Systems Laboratory, 2017).

### 3.3.2 Isoprene SOA formation

GECKO-A treats SOA formation through a dynamic approach that converges towards the equilibrium defined by the Pankow formulation of Raoult's Law (Pankow, 1994). However it is likely that isoprene SOA (ISOPSOA) formation is not only controlled by vapor pressure (Paulot et al., 2009). Among factors that have been identified to play a role in ISOPSOA are: aqueous phase oxidation in deliquescent aerosol (*e.g.* Blando and Turpin, 2000; Ervens et al., 2011; Daumit et al., 2016), organic sulfate/nitrate formation via interaction with the inorganic component of the aerosol (*e.g.* McNeill et al., 2012; Pratt et al., 2013; Wang et al., 2018; Glasius et al., 2018; Jo et al., 2019), and accretion reactions in the bulk aerosol (*e.g.* oligomerization, dimerization, Altieri et al., 2006; Liu et al., 2012; Renard et al., 2015). None of these processes is currently implemented in the GECKO-A framework. For this study we use a simplified approach based on Marais et al. (2016) allowing the representation of ISOPSOA formation depending on the assumed composition of the inorganic aerosol. This parameterization describes the heterogeneous reactive uptake of important isoprene oxidation products. This accounts for the diffusion of the gases to the surface of the wet aerosol particle, their accomodation to the surface and their dissolution. The relevant parameters used here are listed in Marais et al. (2016). Isoprene epoxides (epoxydiols and hydroxyepoxides) react in the aqueous phase to open their epoxide ring via acid-catalyzed reactions. These reactions are followed by either the nucleophilic addition of (i) $H_2O$ to form methyltetrols or (ii) sulfate and nitrate ions to form organosulfates and organonitrates. The uptake of epoxides therefore depends on the acidity of particles, as well as their sulfate and nitrate content. These parameters had to be constrained in the model and were deduced from the T3 AMS measurements and literature data (see Table 1). On the other hand, isoprene oxidation products containing nitrate moieties (dihydroxydinitrates and isoprene nitrate) hydrolyze and form polyols and nitric acid.

## 3.4 Dry Deposition

Dry deposition is treated following the Wesely (1989) parameterization. This parameterization is a resistance model that allows calculating dry deposition velocities based on multiple resistances defined as properties of the surfaces. The city and the forest were respectively attributed the properties of surfaces defined as urban and deciduous forest in the Wesely (1989) paper. The dry deposition velocity of a given species depends on its solubility expressed by its Henry's law coefficient. Because the solubility of most organic compounds generated with GECKO-A is unknown, they are here estimated using the GROupcontribution Method for Henry's law Estimate SAR (Raventos-Duran et al., 2010).

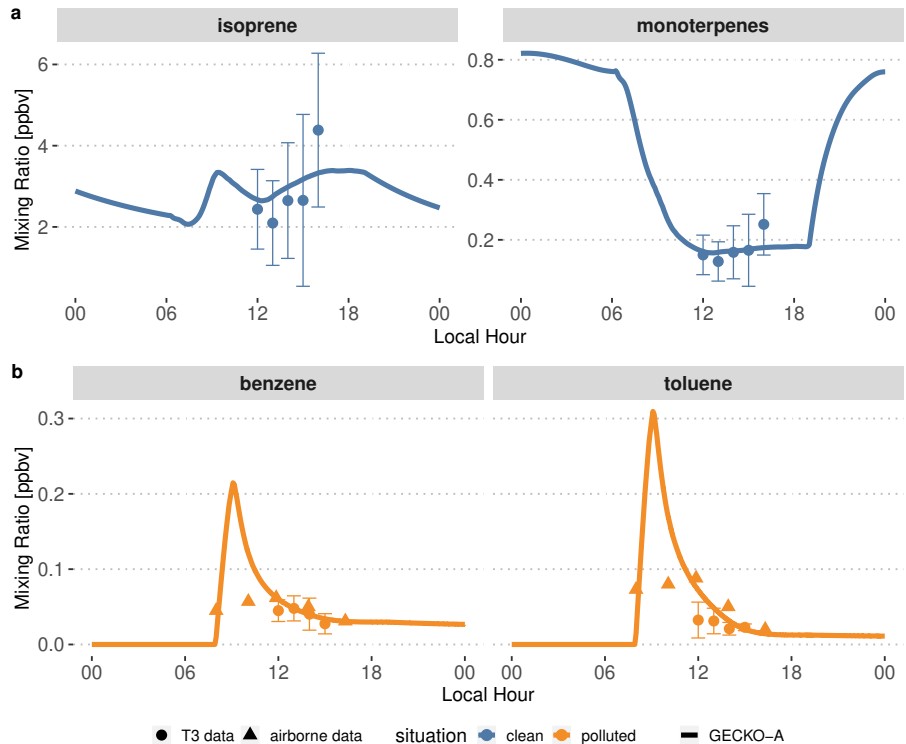

**Figure 5.** Modeled (lines, second day) time evolution of primary species concentrations in the Lagrangian box-model described in Sect. 3.1, average experimental concentrations measured at the T3 site (dots) and in the airplane (triangles). The vertical range of the experimental data denotes the standard deviation of measured concentrations during events identified as clean (top, blue) and polluted (bottom, orange). The airborne data was measured during plume transects. For each transect, aircraft distance from Manaus was converted to a time separation from Manaus assuming the plume leaves the city at 8am and arrives above T3 at 2pm.

## 4 Results and Discussion

### 4.1 Gas Phase Organics: Primary Organic Compounds and Oxidants

Figure 5 depicts the time evolution of selected primary organic species, and compares the model with available measurements. In the clean situations, measured isoprene mixing ratios range from 2–3 ppb at noon to 5–6 ppb at the end of the afternoon. The sum of all monoterpenes follows a similar increasing trend in the afternoon, from 0.1 to 0.3 ppb. After adjusting biogenic emissions rates (see Sect. 3.2.1), the model is able to reproduce these mixing ratios, with isoprene and monoterpenes being simulated to the average of experimental values. In polluted situations, the model shows a peak of anthropogenic organic compounds when the plume encounters Manaus emissions between 8 and 9 am. This peak reaches 0.2 ppb and 0.3 ppb respectively for benzene and toluene (Fig. 5). Their levels decay for the remainder of the day. Because the T3 measurement site is situated 4 to 6 hours downwind of Manaus, measurements of benzene and toluene can be compared to decayed modeled

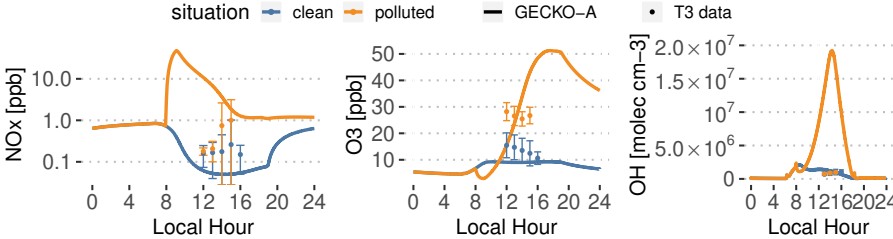

**Figure 6.** Experimental (dots, T3 site) and modeled (lines, second day) time evolution of $NO_x$ (left, note log scale), ozone (middle) mixing ratios and OH radicals concentrations (right). The vertical range of the experimental data denotes the standard deviation of measured concentrations at T3 during events identified as clean (blue) and polluted (orange).

levels after that time span. The modeled mixing ratio of benzene matches the measurements, between 0.4 and 0.6 ppb, while modeled toluene is closer to the higher range of measurements, between 0.2 and 0.6 ppb during the afternoon. Figure 5 also displays airborne measurements of the same anthropogenic compounds during plume transects. The modeled mixing ratios of benzene and toluene decay in a similar way to the concentrations measured at each plume transects. The modeled peak is not seen by the aircraft measurements as the aircraft may not be flying close enough to the emission sources to capture it.

Pristine forest conditions are characterized in the model by low $NO_x$ emissions from the soil ($8.3 \times 10^9$ molec cm$^{-2}$ s$^{-1}$ $\approx$ $1.5 \times 10^{-5}$g m$^{-2}$ h$^{-1}$, see Table 1). The model predicts $NO_x$ mixing ratios around 50 ppt in the afternoon. In the polluted case, the background air mass is exposed to a complex mixture of anthropogenic compounds emissions as well as three orders of magnitude higher $NO_x$ emissions ($\approx 1 \times 10^{-2}$g m$^{-2}$ h$^{-1}$, see Fig. 4). This leads to modeled $NO_x$ around 1 ppb in the afternoon, after a 48 ppb peak in the city in the morning. The increase in $NO_x$ is not as important in the experimental data, but these $NO_x$ measurements are highly uncertain, which could explain the modeled discrepancies.

Daytime ozone mixing ratios are modeled around 9 ppb in the clean situation, in the lower range of measured values. The higher $NO_x$ levels result in strong ozone production in the polluted plume, characterized by mixing ratios of 15 ppb at noon and up to 51 ppb at the end of the afternoon. During this increase of ozone production, the model matches T3 measurements around around 23 ppb at 1pm. On average, measured ozone in the polluted case is a factor of 2 higher than the clean case while the model sees an increase by a factor of 2 to 4 between noon and 6pm. It should also be noticed that the model completely separates clean and polluted situation, which increases the contrast for all variables compared to the classification of the measurements that always includes some degree of mixing (see Sect. 2). It should also be noted that the nighttime decay of ozone can be explained by dry deposition to the forest surface.

Furthermore, VOCs in the plume are exposed to high OH concentrations, with modeled concentration reaching $1.9 \times 10^7$ molec cm$^{-3}$ in the afternoon. In the clean background, OH concentrations only reach $2 \times 10^6$ molec cm$^{-3}$. These clean values are in the lower range of reported measurements at T3 Unlike the model, OH measurements averaged at T3 and identified as clean and polluted did not exhibit any difference between both situations (Fig. 6). In that case, there could be issues with the OH measurements at T3. Indirect constraints have shown differences between clean and polluted situations. Liu et al. (2018) derived

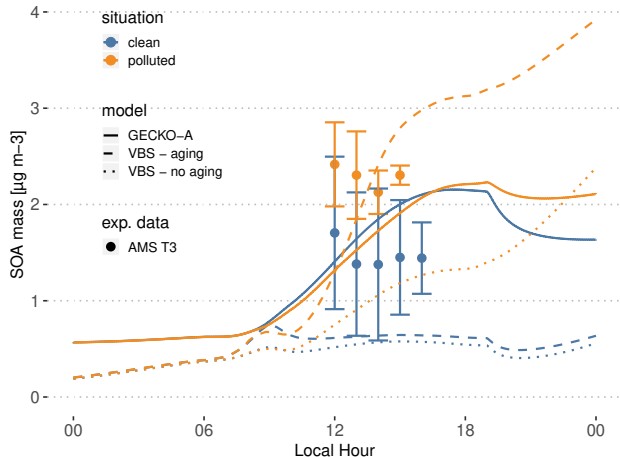

**Figure 7.** Experimental (circles, T3 site) and modeled (lines, second day) time evolution of SOA mass concentration. The vertical range of the experimental data denotes the standard deviation of measured concentrations. Cases are identified as clean (blue) and polluted (orange). The continuous lines depict the GECKO-A model run and the dashed lines depict the modeled SOA mass predicted with the VBS approach from Shrivastava et al. (2019). The dotted lines depict modeled SOA mass predicted with the VBS approach without including aging processes (see Sect. 4.3).

OH concentrations from isoprene and its oxidation products measurement. They showed that noontime OH concentrations vary between $5\times10^5$ molec cm$^{-3}$ in clean situations to $1.5\times10^6$ molec cm$^{-3}$ in polluted events. The Shrivastava et al. (2019) 3D model exhibits a similar OH behavior to this work with concentrations at T3 ranging from $2\sim5\times10^5$ molec cm$^{-3}$ (clean) to more than $4\times10^6$ molec cm$^{-3}$ (polluted). The GECKO-A model is therefore likely to be overestimating OH concentrations in the urban plume by a factor of 5 to 10. This could stem from either overestimating NO or underestimating VOCs emissions in the city.

## 4.2 Modeled Urban Impact on SOA Mass and Composition

### 4.2.1 Modeled vs Measured SOA Mass Concentrations

At the measurement site, SOA mass concentrations measured by AMS range from 0.6 to 2.5 μg m$^{-3}$ in clean conditions. In polluted conditions, SOA mass concentrations range from 1.9 to 2.9 μg m$^{-3}$ (Fig. 7). In the clean case, the modeled SOA mass is within the range of T3 measurements, increasing from 0.6 μg m$^{-3}$ at sunrise to 2.16 μg m$^{-3}$ at the end of the afternoon. In the polluted situation, modeled SOA mass concentration is very similar to the clean simulation, with only a 20 minutes delay in the start of SOA production. The maximum concentration is 2.23 μg m$^{-3}$, only a 3.5% increase compared to the clean simulation, while experimentally this increase averaged around 56%. Because the model is unable reproduce the observed urban SOA enhancement, in the polluted situation the model underestimates SOA mass by 10 to 45%.

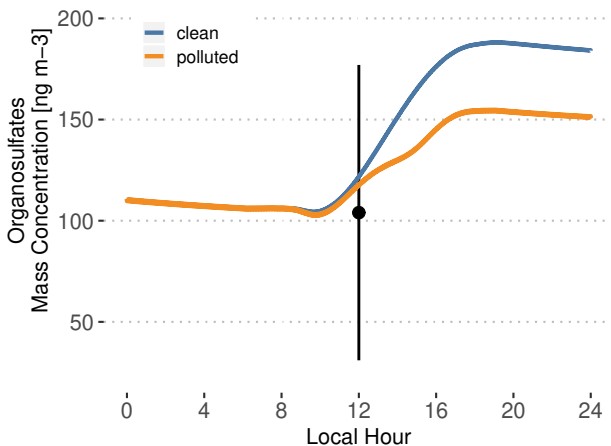

**Figure 8.** Modeled time evolution of particle phase organosulfates mass concentration. Cases are identified as clean (blue) and polluted (orange). The point and vertical line depicts the average and standard deviation of measurements reported in Glasius et al. (2018) for the wet season.

### 4.2.2 Organosulfates

Figure 8 depicts modeled particle phase organosulfates, with mass concentrations ranging from 104 ng m$^{-3}$ in the morning to 188 ng m$^{-3}$ in the evening in the clean case scenario. The polluted situation decreases late afternoon concentrations to 155 ng m$^{-3}$. These values are in the higher range of the reported measured range of 104$\pm$73 ng m$^{-3}$ in Glasius et al. (2018). This is consistent with Glasius et al. (2018) who reported that the main source of the measured organosulfates is IEPOX heterogeneous uptake, which is the only pathway represented in this model. Furthermore, this shows that the combination of the MCM 3.3.1 isoprene oxidation mechanism to produce IEPOX and the reactive uptake parameterization from Marais et al. (2016) is able to predict realistic levels of organosulfates, assuming that aerosol properties are also realistic (hygroscopicity, inorganic sulfates and pH).

### 4.2.3 Modeled Organic Functional Groups

Figure 9 depicts the distribution of organic functional groups in the particle phase. In the clean case scenario, total functionalization, defined as the number of functional groups per carbon atom, is constant around approximately 0.5. As expected for a low-NO$_x$ situation, approximately 40% of these functional groups are hydroxy moieties and 30% of the organic functional groups are hydroperoxides. The remaining functional groups are dominated by carbonyls and nitrates to a lower extent. Manaus pollution has the direct effect of reducing total functionalization by 10% because of the contribution of long-chain primary hydrocarbons to SOA formation in the plume. Oxidation of organics in the higher NO$_x$ environment also leads to an increase of nitrate moieties contribution at the expense of hydroxy and hydroperoxide moieties.

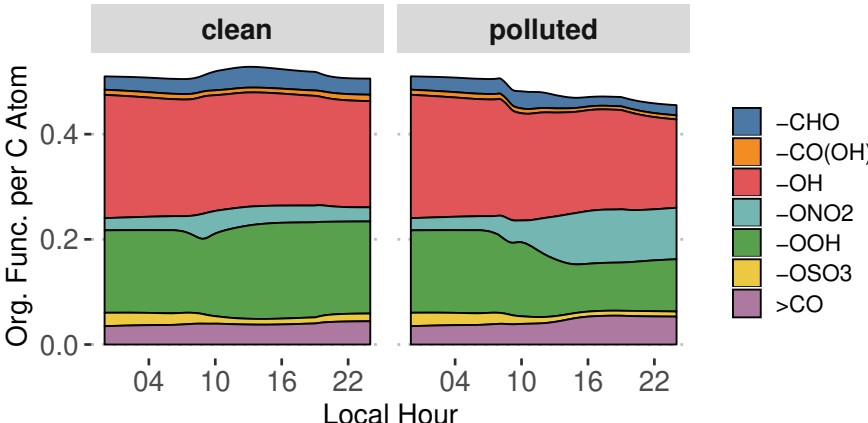

**Figure 9.** GECKO-A modeled time evolution of particle phase organic functionalization for the clean (left panel) and the polluted (right panel) cases. Functional groups are abbreviated as follows: aldehyde (-CHO), carboxylic acid (-CO(OH)), hydroxy (-OH), nitrate (-ONO2), hydroperoxide (-OOH), sulfate (-OSO3) and ketone (>CO). The y-axis is read as the number of a given organic function per carbon atom, i.e. in the clean case there is in total approximately one organic function for every two carbon atom.

The change in overall modeled SOA composition between clean and polluted cases is quite small. AMS measurements give a similar impression of a small impact of polluted situations on atomic ratios (Fig. 10), with only a slight increase of O/C ratio (see Sect. 4.2.4). Other analyses of airborne and ground AMS data (de Sá et al., 2018; Shilling et al., 2018) similarly show that the relative contribution of hydrocarbon-like organic aerosol (HOA) slightly increases in the polluted plume at the expense of isoprene derived SOA. The model and the AMS data support the idea that the impact of anthropogenic emissions is mostly seen on the total organic aerosol mass, and that all constituents of the organic aerosol phase increase approximately in the same way.

### 4.2.4 Modeled vs Measured Atomic Ratios

Figure 10 depicts simulated, ground measurements and airborne measurements of O/C and H/C atomic ratios in aerosol particles on a van Krevelen diagram. At the T3 site, experimental O/C ratios range from 0.7 to 1 in both clean and polluted conditions while H/C ratios range from 1.2 to 1.4. Additionally airborne measurements above the T3 site report O/C ratios ranging from 0.35 to 0.9 and H/C ratios ranging from 1.5 to 1.9. Compiling multiple field campaigns AMS measurements, Chen et al. (2015b) reported van Krevelen diagrams slopes (H/C vs O/C) ranging from -1 to -0.7. A linear regression over the data points from both airborne and ground measurements (dotted line on Fig. 10) gives a slope of -1.3, close to values reported in Chen et al. (2015b). This means that T3 air masses may be sampled at a later stage of oxidation than the airborne samples, possibly because they were exposed to higher levels of oxidants than the higher altitude air masses.

The modeled average particle phase O/C ratios range from 0.77 to 0.86, within the ratios measured at the T3 site. Modeled H/C ratios are however overestimated compared to T3 site measurements, ranging from 1.89 to 1.94. Claflin and Ziemann

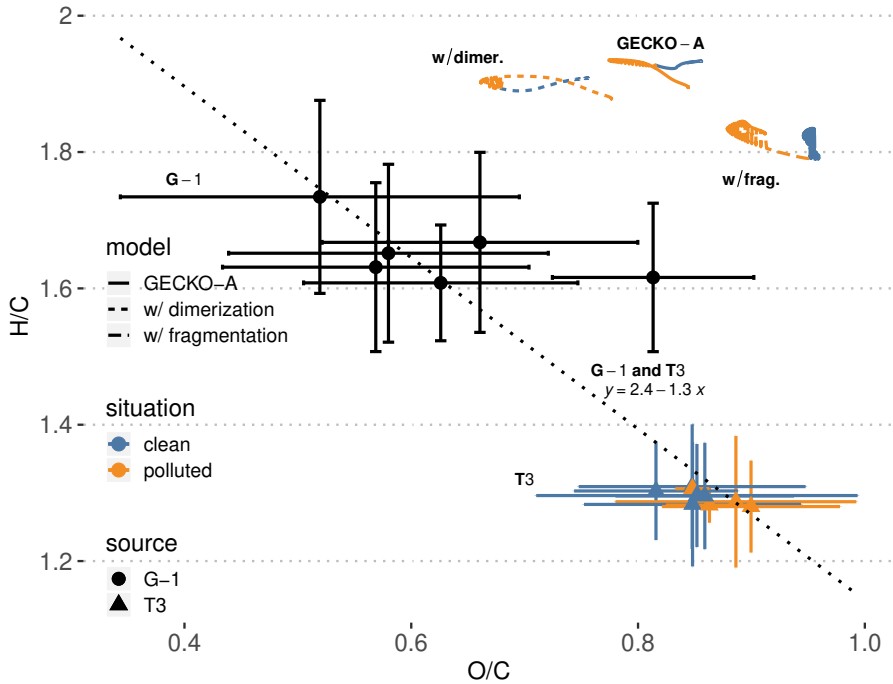

**Figure 10.** T3 site (colored triangles), airborne (black dots) and modeled (lines, afternoon of second day) van Krevelen diagrams of H/C (y-axis) vs O/C (x-axis) average ratios in SOA. The vertical and horizontal range of the experimental data denotes the standard deviation of measured concentrations. Cases are identified as clean (blue) and polluted (orange). Airborne data were filtered to only include measurements taken within 20 km of the T3 site. The dotted line and the associated equation depict the linear regression obtained with all experimental points (T3 and G-1). Modeled lines depict three different calculations (see Sect. 4.2.4): the reference calculation (continuous lines, labeled GECKO-A), a calculation where all $C_{10}$ are supposed to be dimerized (short dashes, labeled w/ dimer.) and a calcualtion where all $C_{10}$ are supposed to fragment (long dashes, labeled w/ frag.)

(2018) reported experimental evidence that the reaction of $\beta$-pinene with $NO_3$ produces oligomers derived from $\beta$-pinene $C_{10}$ oxidation products. For instance one of the proposed mechanisms for dimerization of a $C_{10}H_{17}O_5$ (H/C = 1.7) produces

a $C_{20}H_{30}O_9$ (H/C = 1.5). In the GECKO-A modeled aerosol phase, after organosulfate and nitrates derived from isoprene, $C_{10}$ compounds dominate OA composition. As examples, a $C_{10}H_{20}O_6$ (H/C = 2; O/C = 0.6) and a $C_{10}H_{18}O_7$ (H/C = 1.8; O/C = 0.7) derived from limonene are the second and third most important organic species in the aerosol phase on a molecule basis. Following the dimerization pathways suggested by Claflin and Ziemann (2018), these compounds could potentially form $C_{20}H_{36}O_{11}$ (H/C = 1.8; O/C = 0.55) and $C_{20}H_{32}O_{13}$ (H/C = 1.6; O/C = 0.65) dimers respectively. Dimerization, or similar

oligomerization processes, would then possibly move the modeled van Krevelen diagram towards lower H/C ratios, closer to AMS measurements.

As a test, we generalized this estimation to all $C_{10}$ in the aerosol phase: we replaced each $C_{10}$ by the corresponding $C_{20}$ and halved its concentration. In this way, we can calculate what would H/C and O/C ratios be in the aerosol phase if aging processes

only dimerized $C_{10}$ compounds. The resulting modeled van Krevelen diagram is reported on Fig. 10 (labeled w/ dimer.). The impact of $C_{10}$ dimerization is relatively strong on O/C ratio, ranging from 0.66 to 0.78 and remaining in the range of measured O/C ratios at T3 site and in the aircraft. H/C ratios are only reduced to 1.88–1.94, still 50% higher than measured H/C at the T3 site and 20% higher than airborne data.

Oppositely, GECKO-A could be missing processes that would fragment the aforementioned two $C_{10}$ compounds. Fragmenting $C_{10}H_{18}O_7$ into a $C_4H_6O_4$ (H/C = 1.5; O/C = 1) and a $C_6H_{10}O_5$ (H/C = 1.7; O/C = 0.8) species would bring the average H/C ratio down from 1.8 to 1.6. This possibility of missing fragmentation processing means that either the modeled gas phase chemistry doesn't compete enough with condensation to fragment these species, or these $C_{10}$ species should be fragmented by heterogeneous or condensed phase processes in the particles themselves, which are not accounted for by the model. It should be noted that because the fragmented compounds are lighter, they would exhibit higher volatility. However this does not necessarily mean that the SOA mass would decrease because these shorter species are still oxygenated, maybe enough to contribute to SOA mass through solubility controlled processes in the same fashion as what is known about isoprene oxidation products.

As another test, we also estimated what would O/C and H/C ratios be if all $C_{10}$ fragmented in the aerosol phase. The resulting modeled van Krevelen diagram is reported on Fig. 10 (labeled w/ frag.). In this case, modeled O/C ratios increase to a range of 0.88 to 0.96 and remain in the higher end of measured ratio at the T3 site. H/C are reduced further than in the dimerization test and sit at the higher end of airborne measured H/C ratios, but they still are 45% higher than H/C ratios measured at the T3 site.

Even if they apparently cannot account for the discrepancy between modeled and measured H/C ratios, the two tests presented here on $C_{10}$ compounds in the aerosol phase show the potential importance of adding these missing processes in GECKO-A. These simple tests are however simplifications that overlook important factors in the potential impact on SOA composition: (i) not all $C_{10}$ compounds would be affected by these processes, (ii) other compounds than $C_{10}$ could react in a similar way, (iii) trimerization, tetramerization and other accretion processes could also occur in the aerosol phase, (iv) missing fragmentation processes could also happen in the gas phase.

### 4.3 Comparison with VBS approach

Shrivastava et al. (2019) modeled this same field campaign with WRF-Chem, a chemistry transport regional model (Grell et al., 2005) and similarly to this work they based their primary organic compounds emissions on the MEGAN inventory (Guenther et al., 2012) for biogenic compounds, and the methodology described in Andrade et al. (2015) and data from Medeiros et al. (2017) for anthropogenic emissions. Using a Volatility Basis Set approach (VBS) to account for condensation of low volatility species, and considering ISOPSOA separately with an approach similar to this work, they modeled airborne SOA mass to within 15% of airborne measurements. The VBS parameterization described in Shrivastava et al. (2019) represents the formation of SOA as four surrogate species differing by their volatility ($C^\star$ = 0.1, 1, 10 and 100 µg m$^{-3}$). For biogenic SOA, isoprene and monoterpenes produce these four surrogates from the oxidation by OH, ozone and $NO_3$, with yields depending on $NO_x$. Moreover multigenerational aging is accounted for the surrogate species assigning fragmentation (*i.e.* increasing volatility) and functionalization (*i.e.* decreasing volatility). This aging is parameterized as a reaction of each of the SOA surrogate species

VBS$_n$ with OH as follows:

$$VBS_n + OH \rightarrow \alpha_{\mathrm{frag}}VBS_{n+1} + (1 - \alpha_{\mathrm{frag}})VBS_{n-1} \tag{R1}$$

The reaction rate is $k_{R1} = 2 \times 10^{-11}$ cm$^3$ molec$^{-1}$ s$^{-1}$. The branching ratio for fragmentation $\alpha_{\mathrm{frag}}$ is determined as the ratio
of the reaction rate of peroxy radicals with NO to the sum of all peroxy radical reactions rates; it has an upper limit of 75%.
The yields used in this VBS approach were fitted over a variety of low OA loading atmospheric chamber studies of biogenics
oxidation under high and low NO$_x$ concentrations (Shrivastava et al., 2019). More details about this VBS approach can be
found in Shrivastava et al. (2013, 2015, 2019).

In order to compare the GECKO-A model results with the VBS approach used in Shrivastava et al. (2019), additional
simulations were run where the explicit condensation of low volatility biogenic species was replaced by the formation of the
four surrogate species used in Shrivastava et al. (2019). Fig. 7 shows the time evolution of predicted SOA mass with GECKO-
A, after replacing the original condensation of low volatility biogenic species by the VBS approach used in Shrivastava et al.
(2019) (dashed lines). In this test, the VBS modeled SOA mass is well within the range of measured values in the afternoon
for the polluted case scenario. The VBS version of the box-model is however underestimating SOA mass concentrations in the
clean situation, with only 0.5 µg m$^{-3}$ during daytime compared to the measured 0.6 to 2.5 µg m$^{-3}$ range. Like in Shrivastava
et al. (2019), exposure of the background air mass to the urban increased oxidative capacity increases VBS predicted SOA mass
by almost 400%, which explains how the VBS can reach the higher polluted case SOA mass. Figure 7 also depicts the predicted
SOA mass if SOA aging is not included in the VBS model (dotted lines). Shrivastava et al. (2019) reported that SOA aging does
not have a strong effect in their simulations, which is not the case when applied in the box-model. In our simulation without
aging processes, the polluted case SOA mass concentration drops below 1.3 µg m$^{-3}$ in the afternoon. However in the clean
case scenario, the SOA mass concentration only decreases by approximately 10% when SOA aging is removed. This means
that SOA aging becomes more important in the ground case scenario when the air mass is exposed to high OH concentrations
that were not seen by the model run by Shrivastava et al. (2019): their maximum OH concentrations reach $2 \times 10^6$ molec cm$^{-3}$
while our maximum OH concentration reach $1.6 \times 10^7$ molec cm$^{-3}$.

Figure 11 and Table 2 attribute sources of SOA according to the GECKO-A explicit simulation and the VBS approach. In the
clean case scenario, GECKO-A attributes most of SOA mass to monoterpene oxidation products (65% at 2pm). The remainder
is attributed to isoprene SOA, with condensation of low volatility compounds contributing in the same proportion as reactive
uptake (17% and 18% respectively). In Shrivastava et al. (2019), monoterpene oxidation products account for 45% of SOA
sources in the airborne plume. With their VBS applied to the ground situation, 28% of SOA is attributed to monoterpenes at
2pm, approximately half of the proportion predicted by the GECKO-A explicit approach. Like in the 3D model calculation,
the VBS in the box model attributes the remainder of background SOA mass mostly to reactive uptake of isoprene oxidation
products (53% of total SOA).

In the polluted case, the explicit model predicts a slight decrease of 6% in total SOA at 2pm while measurements exhibit
an increase of 33% on average. The urban effect is strong in the VBS case than the explicit approach with a 380% increase
in mass. In the comparison with airborne measurements, the Shrivastava et al. (2019) model predicts that the city oxidants

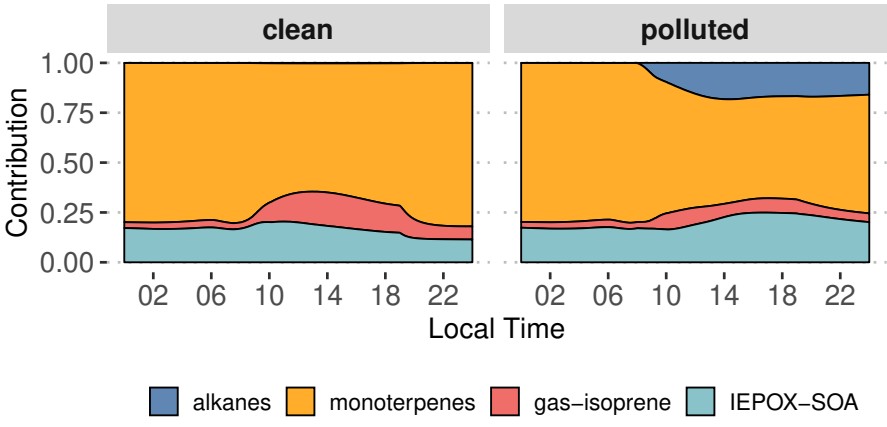

**Figure 11.** Contribution of primary hydrocarbons categories to GECKO-A modeled SOA mass for the clean (left panel) and polluted case (right panel).

**Table 2.** Contribution of primary hydrocarbons categories to modeled SOA mass at 2pm. Percentages in parentheses indicate the relative contribution to total SOA mass.

| SOA mass [$\mu$g m$^{-3}$] | GECKO-A clean | GECKO-A polluted | VBS - aging clean | VBS - aging polluted | VBS - no aging clean | VBS - no aging polluted | Measured[a] | |
|---|---|---|---|---|---|---|---|---|
| Monoterpenes | 1.19 (65%) | 0.91 (53%) | 0.18 (28%) | 0.71 (30%) | 0.14 (24%) | 0.17 (16%) | – | – |
| Isoprene (gas) | 0.31 (17%) | 0.11 (6%) | 0.12 (19%) | 1.00 (41%) | 0.09 (16%) | 0.18 (17%) | – | – |
| IEPOX-SOA | 0.34 (18%) | 0.39 (23%) | 0.34 (53%) | 0.39 (16%) | 0.34 (60%) | 0.39 (37%) | – | – |
| biogenics | 1.84 (100%) | 1.41 (82%) | 0.64 (100%) | 2.1 (87%) | 0.57 (100%) | 0.74(70%) | – | – |
| anthropogenics | 0 (0%) | 0.32 (18%) | 0 (0%) | 0.32 (13%) | 0 (0%) | 0.32 (30%) | – | – |
| total | 1.84 | 1.73 | 0.64 | 2.42 | 0.57 | 1.06 | 1.4$\pm$0.8 | 2.1$\pm$0.2 |

[a]de Sá et al. (2018)

cause the same large increase in biogenic SOA formation (up to 400%), and that this increase is due to enhanced monoterpene oxidation. With GECKO-A at the ground site, SOA mass remains constant because of the contribution of anthropogenics which compensates the decrease in the contribution from the condensation of isoprene and monoterpenes oxidation products by 32%. This loss is slightly compensated for by an increase in the production of SOA via reactive uptake of isoprene oxidation products (15% increase) because the plume favors these processes with higher sulfate load and lower pH (see Table 1). Overall biogenic SOA decreases by 23% with respect to the clean case. In the VBS test case, SOA mass formed from condensation of low volatility oxidation products of isoprene and monoterpenes is enhanced in the polluted case respectively by a factor of 7 and 3. This enhancement is notably inhibited when the aging parameterization is removed from the VBS approach with a mass

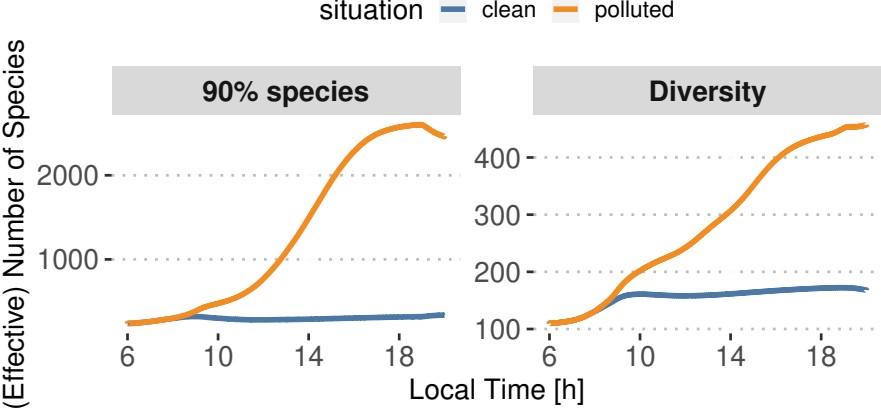

**Figure 12.** Smallest number of species needed to capture 90% of modeled SOA mass (left panel) with GECKO-A at each timestep ($N_{90\%}$, see text) and statistical diversity $D$ in the GECKO-A modeled particle phase (right panel, see Eq. 3).

increase due to condensation of low volatility products of isoprene and monoterpenes of respectively 100% and 21%. This
highlights the importance of modeling aging of low volatility oxidation products to explain the enhanced production of SOA in the urban plume.

### 4.4   Potential for Reduction of the Explicit GECKO-A mechanism

It is obvious that the chemical mechanisms generated with GECKO-A are too large to be implemented in 3D models. The GECKO-A mechanisms need to be reduced to sizes manageable by 3D models, typically a few hundred species and reactions.
The VBS parameterization used for comparison in this work is fit for low OA loadings, biogenic dominated situations but it is unclear that it should be applied to other situations.

    In this section, we are *not* proposing a much needed new approach to reducing explicit mechanisms with the goal of predicting SOA mass concentrations, but we explore here the potential for reduction of the chemical mechanism that was generated for this study. In other words, what is the theoretical lower limit to the number of species that should be used in a reduced
scheme to still be able to model the same SOA mass concentration time profile as the explicit model?

    To answer this, two metrics are presented in Fig. 12. The first one $N_{90\%}$ is the smallest number of species needed in the explicit model to capture 90% of the total SOA mass at each timestep. After sorting species by decreasing concentration, this number is calculated by adding up these concentrations until 90% of the total modeled SOA mass is reached. The operation is repeated at each timestep. Calculated independantly, the second one is the particle diversity $D$ in the explicitly modeled SOA,
as defined for instance in Riemer and West (2013):

$$D = \exp S \tag{3}$$

where $S$ is the first order generalized entropy (also known as Shannon entropy):

$$S = \sum_{i=1}^{N} -p_i \ln p_i \qquad (4)$$

where $p_i$ is the mass fraction of species $i$ in the organic particle phase and $N$ is the total number of species in the organic particle phase. As stated in Riemer and West (2013), the diversity is a measure of the effective number of species with the same concentration in the organic fraction of the aerosol phase. If $D = 1$, the organic fraction is pure as it is composed of a single species. Therefore, a value $D \ll N$ means that of all the species contributing to modeled organic aerosol, only a few significantly contribute to its composition. Oppositely, $D = N$ is the maximum value reachable by $D$ and is obtained when the organic fraction is composed of $N$ equally distributed species. In the case where $D$ is close to $N$, only a few species are negligible. For more details and better explanations, we refer the reader to Riemer and West (2013, esp. Fig. 1). We make the hypothesis here that $D$ can be interpreted as an effective number of species derived from the informational entropy of the modeled particle phase.

In the clean situation both metrics behave similarly, with a morning increase of the number of species until 10 am, after which the number remains relatively constant until sunset. During daytime, on average $N_{90\%} = 292$ species are needed to represent 90% of the SOA mass. The calculated diversity is around 153 effective species. For the polluted situation, $N_{90\%}$ increases during daytime by about a factor of 9, reaching about 2500. The calculated diversity only increases up to approximately 260 effective species. These increases in the species numbers for the polluted case are logical as the variety of precursors, and hence secondary species that could potentially contribute to SOA, is increased by urban emissions.

The number of species needed to represent most of the modeled SOA mass in all cases seems too high to be used in 3D models applications. Furthermore there is no guarantee that the most important species at a given timestep would be the same most important species at the following timestep. This suggests that reductions should not come from simply selecting species identified as important to represent the variety of species that could arise in the interaction of biogenic air and an urban plume.

The statistical diversity calculation seems like a better approach to estimate the minimum number of species needed to model SOA mass. As this number is directly derived from informational entropy, we suggest that the diversity represents the number of species that would be needed to reproduce the same informational content regarding the time evolution of SOA mass in the explicit model. Even if the effective species numbers fall in the higher range of what would be acceptable in a 3D model chemical mechanism, the practical construction of the mechanism remains to be explored. For instance, in the polluted scenario, $D$ is a factor of 9 lower than $N_{90\%}$. This should mean that $D$ cannot represent a subset of the individual species from the original mechanism, otherwise it would be expected to be equal or higher than $N_{90\%}$ if it is supposed to reproduce the informational content regarding SOA mass. It is therefore likely, and making this problem more complex, that each of these effective species is a (non) linear combination of explicit individual species.

Finally, we used in this section an entropy calculation for SOA mass: it is based only on mass fractions of the species composing the modeled organic particles. The effective number of species displayed on Fig. 12 is therefore only meaningful for SOA mass and properties directly linked to it. If the goal is to predict other properties, *e.g.* hygroscopicity, toxicity or optical properties, assuming we find a way to calculate these with GECKO-A, the diversity defined here would not necessarily

be meaningful. For instance, hygroscopicity or toxicity could be driven by a handful of oxygenated species that do not matter for the informational content regarding SOA mass. We did not explore further down this path, as this is not the subject of this paper, but it may be possible to generalize this definition of informational diversity to properties other than mass.

## 5   Conclusions

An explicit chemical mechanism generated with GECKO-A was used in a box model to simulate a situation similar to the situation studied in Manaus during the GoAmazon 2014/5 field campaign. After scaling down the emissions generated from the MEGAN biogenic emissions model and estimating urban emissions in Manaus, the model was able to reproduce realistic primary organic compounds mixing ratios as well as $NO_x$, ozone and OH concentrations.

The model is able to reproduce background SOA mass concentrations but is not able to reproduce the observed enhancement 490 in the polluted plume. When running a Volatility Basis Set approach that was previously applied to the Manaus case (Shrivastava et al., 2019), modeled SOA mass matches measurements which suggests that the incorrect explicit model prediction is not caused by incorrect primary organic compound emissions or oxidant levels. Modeled particle phase organosulfates are within the range of previous measurements (Glasius et al., 2018) which suggests that isoprene oxidation and SOA formation in the model are reasonably well simulated. In both polluted and clean situations, biogenics are identified as the main con-495 tributors to SOA by both GECKO-A and the VBS parameterization. In both approaches, the majority of SOA production is attributed to monoterpenes oxidation and condensation of lower volatility products. Yee et al. (2018) measured and described sesquiterpenes during GoAmazon 2014/5 for the same situations and suggested that these species may be important for modeling studies. However the modeling study of Shrivastava et al. (2019) estimated that the contribution of sesquiterpenes to SOA production is less than 10%. It is more likely that physico-chemical processes involved in monoterpene SOA formation are ei-500 ther unknown or missing in the explicit model. Comparison of modeled and measured elemental ratios (H/C and O/C) indicates that fragmentation of monoterpenes oxidation products and their condensation or reactive uptake to the condensed phase may play an important role in understanding the sources of biogenic SOA mass. This reactive uptake may in turn involve oligomerization and fragmentation processes. However, simple sensitivity tests show that these processes alone may not explain the discrepancies between the explicit model and measurements. Because the VBS parameterization is based on multiple chamber 505 experiments, it could implicitly be accounting for these missing processes. Of the high diversity of monoterpenes identified in Amazonia (Jardine et al., 2015), only a handful of monoterpenes have been studied to the extent that we can be as confident in model predictions of SOA formation from monoterpenes as from isoprene. Detailed mechanistic studies of monoterpene oxidation are therefore needed for further incorporation in explicit models to better understand the nature and the magnitude of the contribution of monoterpenes to SOA formation, as well as their response to the interaction with urban pollution (*e.g.* 510 Claflin and Ziemann, 2018).

Even if a parameterization was implemented in GECKO-A to properly address the formation of isoprene SOA via aqueous phase processes (Marais et al., 2016), to explicitly treat these in a more general way, future GECKO-A developments for mechanism generation will need to include the following: (i) aerosol thermodynamics, for instance via coupling with a model

like MOSAIC (Zaveri et al., 2008) or ISORROPIA (Nenes et al., 1998), (ii) aqueous phase processes including explicit disso-lution (*e.g.* Mouchel-Vallon et al., 2013), oxidation (*e.g.* Mouchel-Vallon et al., 2017), accretion reactions (*e.g.* Renard et al., 2015), and interaction with dissolved inorganic ions, (iii) explicit treatment of the fate of newly formed species like dimers or organo-sulfates.

One could be tempted to think that since the VBS parameterization is behaving particularly well in this GoAmazon 2014/5 case, it could be the answer to predict SOA mass in larger scale 3D models. However this approach is limited by the fact that it was fitted for low biogenic OA loading situations and was run in a limited domain regional model (Shrivastava et al., 2019). One possible way of building reduced mechanisms is to reduce existing detailed chemical mechanisms to sizes manageable by 3D models (e.g. Szopa et al., 2005; Kaduwela et al., 2015). Using an information theory based approach, we provide here a lower limit to the size of these reduced mechanisms, assuming the goal is to produce the same informational content as the explicit mechanism. This lower limit of a few hundred species is four orders of magnitudes lower than the actual number of species that are actually accounted for in the explicit mechanism ($4\times10^6$) and shows the potential for progress in future mechanism reduction endeavors. Even if a direct application of this statistical approach to create a reduced mechanism would likely require some atmospheric chemistry breakthrough, it could at least currently be used as a statistical indicator for comparing reduced mechanisms with reference explicit mechanisms.

*Code and data availability.* The GoAmazon 2014/5 experimental data is available from the ARM website: https://www.arm.gov/research/campaigns/amf2014goamazon

The chemical mechanism generated for this study is available upon request from CMV in text or netcdf format.

*Author contributions.* CMV, AH, DG, JLJ, DHL and SM conceptualized and created the methodology. PA, JLJ, STM, JN, BBP and JES collected and curated the experimental data. CMV carried out the formal analysis and investigation of the model results with support from AH, MC, MS and SM. SM and BA originally designed the model. CMV and JLT developed and ran the model. SM and AH secured CMV's funding. CMV wrote the original draft. All authors discussed the results and commented on the paper. CMV carried out the review and editing of the paper, with support from all the authors.

*Competing interests.* The authors declare no competing interests.

*Acknowledgements.* The National Center for Atmospheric Research is sponsored by the National Science Foundation. We gratefully acknowledge support from U.S. Department of Energy (DOE) ASR grant DE-SC0016331. JLJ and BBP were supported by NSF AGS-1822664 and EPA 83587701-0. This manuscript has not been reviewed by EPA, and thus no endorsement should be inferred. Dr. Shrivastava was also supported by the U.S. DOE, Office of Science, Office of Biological and Environmental Research through the Early Career Research Program.

Data were obtained from the Atmospheric Radiation Measurement (ARM) User Facility, a U.S. DOE Office of Science user facility managed by the Office of Biological and Environmental Research. The research was conducted under scientific license 001030/2012-4 of the Brazilian National Council for Scientific and Technological Development (CNPq). We are grateful to Louisa K. Emmons for providing the MEGAN emissions data; and Suzane S. de Sà for providing the clustering analysis results. We are thanking Siyuan Wang for helpful comments.

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
