# Peer review of "Exploration of Oxidative Chemistry and Secondary Organic Aerosol Formation in the Amazon during the Wet Season: Explicit Modeling of the Manaus Urban Plume with GECKO-A"

_Atmospheric Chemistry and Physics, 2019_

## Referee Comment (RC1) · Anonymous Referee #1 · 30 Dec 2019

This manuscript describes box model simulations of secondary organic aerosol formation in Amazonia. SOA formation is simulated using the explicit mechanism generator GECKO-A. Results from GECKO-A are compared to SOA predicted using a volatility basis set (VBS) parameterization.

The title of the article includes "Impact of Urban Emissions on a Biogenic Environment," however I feel like those urban impacts are not articulated very well in the current manuscript. The impacts of the urban plume on SOA formation are buried a bit under model details and do not seem to be discussed in great enough detail for readers

to walk away knowing what these urban impacts are. For example, the key figures outlining urban impacts seem to be Figures 7 and 9; neither of these figures is discussed in great detail. In particular, Fig 9 does not receive much attention at all.

Figure 1 - some readers with colorblindness may not be able to distinguish the shades of red, pink, and green used in the maps.

Section 3.1.1 and Figure 2 - Line 112-113 state "The top box extends from the top of the planetary boundary layer to 1.5 km and represents the residual layer (RL)." However the black dashed line in Fig 2 makes it look like the top of the second box is at ~900 m.

First paragraph of 3.2.2 - the vehicle patterns and emissions in Manaus are assumed to be similar as in Sao Paolo. However, as noted in line 163-164, the fuel used in Manaus is very different than in Sao Paolo. There are many papers on how ethanol blends impact vehicle emissions; the authors should at least acknowledge this literature and comment on how their assumptions about vehicle emissions might impact their results.

Fig 5 - the biogenics in panel a and b show 5 hours where the box is over T3, but the anthropogenic panels c and d only show 4 hours over T3. Why is there this difference?

Figure 9 should be discussed more. The compositional differences between the clean and polluted cases seem very small. Is this consistent with AMS data?

I'm unclear on what is plotted in Fig 12b. Is this the overall species diversity from the model? Or is it the number of species needed in the reduced model to reproduce 90% of the diversity from the full GECKO model run?

I'm also confused about Fig 12a. Was the number of species determined by adding up the number of species needed to capture 90% of the SOA mass (i.e., simply doing a mass balance)? Or was the model run in some sort of reduced form?

[Figure]

2019.

---

## Referee Comment (RC2) · Anonymous Referee #2 · 3 Jan 2020

Review of manuscript acp-2019-1024, "Impact of Urban Emissions on a Biogenic Environment during the wet season: Explicit Modeling of the Manaus Plume Organic Chemistry with GECKO-A"

The authors present an application of the GECKO-A model to GoAmazon2014/5 campaign to understand impacts of the Manaus plume on biogenic SOA, with a focus on the T3 site. Overall, I think it is valuable work, and continued investigation and application of the GECKO-A model stands to deepen our understanding of the "molecular view" of the atmosphere. However, I'm not sure the authors fully achieve their stated

goals of improved understanding of the influence of the plume, which is used in this work primarily as a benchmark against which to test the model. Instead, I think their discussions of the model strengths and weaknesses are more enlightening, and an expansion of some of these discussions may further increase the scientific value of this application. In general, this work is important, and fitting for this journal, but I think there are some places that need major revision.

Major comment: (1) The focus on this work claims to be on anthropogenic influence, but one major result is the relative lack of skill GECKO-A has in capturing the impact of the Manaus plume on SOA production. This suggests to me that a major result might actually be just that the box model is poorly including anthropogenic emissions. SOA formation is underpredicted, OH concetnrations are significantly over predicted, so perhaps there are just more or more reactive VOCs in the city than being included in the model. This is further exacerbated by the issue that the surrogate composition of the fuel emissions are known to not really accurately represent the true composition. The ability of the model to capture clean conditions, which with MEGAN and the PTR have reasonably well constrained emissions, speaks well of GECKO-A, and suggests just that the issue may be in the emissions being fed to the model.

(2) This paper is primarily an exploration of the strengths and weakness of GECKO-A, it doesn't do much to discuss the "Impact of Urban Emissions" as the title suggests. There is some discussion, but mostly it is a comparison of GECKO-A to other models (VBS, Shrivastava) and not really providing new information beyond a further exploration of GECKO-A. I think there are interesting results regarding GECKO-A, in particular the discussion of reduced complexity, which seems like a real place for GECKO-A to provide generalizable scientific insight (how complex do we really need models to be?), and that the weaker part of the work is the attempts to understand the Manaus plume (for which the emissions may not be correct, and aging may not be incorporated, and other issues). Potential issues in modeling the plume end up conflated with potential issues in the model, and instead I think perhaps some of the discussions about the

model could instead be bolstered and thought about more deeply (e.g., I don't think the explanations for H/C discrepancies are likely complete).

Technical comments: Line 1. Missing "of" : "investigation of the"

Line 7. Not clear what it means for the model to "reproduce measured primary compounds" after tuning emissions. Aren't the primary compounds just the emissions, so you tune for this result, and it isn't really impacted by the skill of the model? Perhaps it will be more clear to me after a detailed reading.

Line 15. "particularly intense all year long" is a bit odd, perhaps the authors mean "more photochemically active than other regions throughout most of the year"?

Line 36-37. This mid-paragraph question is odd. "Would" in what case? Do the authors mean "Do"? Maybe just rephrase this without the use of this rhetorical question.

Line 50. The discussion of the "molecular view," while valuable, has some gaps or issues here. In particular, those citations of Koss et al. are an odd choice, as I believe they are just using a PTR, and if I'm not mistaken, there was a PTR run at T3 by the Martin group (some of the data of which is used in this work). Other groups (e.g. the Kroll group, and the CLOUD group) have tried to combine multiple spectrometers to actually capture the whole range of compounds, reaching more a molecular view. However, even still, this would be more reasonably considered a "formula view", as these instruments do not separate molecules out from their formulas. There were also several other instruments at T3 approaching a molecular view (I believe for instance the Goldstein group has collected and run GCxGC of filter samples, providing some molecular invormation), though perhaps not comprehensively. I would re-frame some of this discussion to more accurately capture the landscape and discuss what existing measurements can or can't provide (e.g., I agree the available instruments probably don't provide a comprehensive measurement to compare to models, particularly for gas-phase oxygenates).

Line 54. I'm a big fan of GECKO-A, but maybe not everyone will agree that it is "the ideal tool". "an excellent tool" perhaps?

Line 171. The authors discuss the fact that n-alkanes are perhpas not a good surrogate for diesel fuel and gasoline IVOCs, which are mostly branched and cyclic. However, they do not explain why their estimates are less branched than other work has suggested, and importantly they do not disucss the impacts of these structural differences. They acknowledge that almost none of diesel fuel is comprised of the compounds they are using as surrogates, but do not further discuss this issue. Gentner et al. attempt to put estimates on the impact of branching and rings on SOA, so estimating this uncertainty shouldn't be too difficult.

Figure 5. The measurement of benzene and toluene at T3 are fairly poor constraints on these species as they catch only the tail end of the decay (and even still don't really agree with the toluene model). Is there no aircraft data or VOC measurements at T1 or T2 near Manaus to better constrain these?

Line 260. Again, are there T1 or T2 measurements that could help constrain VOCs in the city? Also, to what extent does model include residual biogenic VOCs present in the city? Presumably there are VOCs present in the city other than just the vehicle emissions, like biogenics from the surrounding forest (or volatile chemical products?), are those captured by the model? Is there a spin up time to allow the city emissions to have some equilibrium concentration of VOCs that would help suppress OH concentrations?

Figure 8. It would be helpful to add Glasius OS measurmenets to the figure as a dashed line.

Line 300-314. I'm not sure the explanations provided for the H/C disparity can really close the gap and it warrants further discussion. The examples the authors provide of oligomerization and fragmentation provide relatively modest decreases in H/C and also change the O/C. A huge fraction of the compounds would need to be oligomers or

fragments for this to reconcile H/C, and this would likely shift the O/C. In the examples they provide, dimers still have H/C ratios well above the observed average. If, for instance, all of the C10 compounds were actually dimers, wouldn't this just bring the average down to 1.5 or 1.6? How do you get down to 1.3? What sorts of compounds can push H/C this far down? Trimers? Tetramers? And would the whole mass need to be comprised of these? The authors could play some games with their data to explore this, for instance assume all compounds are actually dimers or fragments and estimate the average H/C and O/C. This might further be scientifically interesting by putting some constraints on accretion products.

Line 316. How do the anthropogenic emissions used by Shrivastava compare to those used here?

Line 322. I don't fully understand the aging parameterization, could the authors provide more detail?

Line 362. The authors point out the importance of aging in capturing the polluted SOA, which the GECKO-A model does not really capture. Is this due to a lack of aging in the GECKO-A model? Maybe I missed it, but is the GECKO-A model only oxidizing the gas phase and not aging the particle? Considering the importance of aging on reproducing the SOA mass (Figure 7), could the authors include a parameterization of aging in the GECKO model? This might also address the H/C and O/C issues.

Line 367-368. Why is it unclear? It seems to capture polluted periods better than biogenic periods (Figure 7).

Line 400-402. The ideas of reducing complexity discussed in this section are very interesting. It's not completely clear to me that some of the conclusions aren't overextended. In particular, the conclusion that "diversity represents the number of species that would be needed to reproduce the same informational content regarding the composition of SOA." The parameter D is based solely on mass fraction, not their physicochemical properties, so it might not capture other properties such as oxygenation or volatility.

Imagine a scenario where a small fraction of highly oxygenated compounds drive up O/C - this might still impact hygroscopicity but would this complexity be captured by D? Or a small fraction of more volatile components might partition between the gas- and particle-phase and drive oxidation of particle mass. While the exact mechanism of implementing such a reduction is out of scope of this mansucript, the overall point is that it's not to me that either of the diversity parameters really capture the complexity. Would it be possible to implement a variant of D for parameters of interest? For instance minimum number of components needed to describe O/C within 10% while capture some fraction of the mass? I'm not sure the best parameters, but the idea would be something that captures some of the properties beyond simply mass.

Line 403-405. Isn't it not only possible but certain that an effective species is not an individual species but rather a combiniation of explicit species? Otherwise D would be more similar to the other metric, or equal to N. The very nature of the parameter D is to be a mathematical descriptor, not an individual species, correct?

---

## Author Comment (AC1) · 23 Mar 2020

**Response to Anonymous Referees**
**Impact of Urban Emissions on a Biogenic Environment during the wet season: Explicit Modeling of the Manaus Plume Organic Chemistry with GECKO-A**

Camille Mouchel-Vallon et al.

We thank both reviewers for their helpful comments. Below is our detailed answer to their specific comments, followed by the marked up manuscript.

**Major Updates**

- During the review process, we discovered an issue with the GECKO-A generated mechanism that cut off oxidation too early. As a result, later generations species were not accounted for in the explicit mechanism. A new mechanism has been generated after fixing this issue and was used to produce new model simulations. Although the contribution of missing species increased SOA mass by 50% in the clean scenario, the predicted values remain in the range of observed SOA mass at the T3 site, and our conclusions remain unchanged.

- In addition, as we explain in the answers to comment 8, we updated the anthropogenic emissions with emissions from the local power plants and oil refinery. The urban emissions of $SO_2$ and $NO_x$ were significantly increased and this has an impact on the modeled urban plume SOA. The tables, figures and text have been updated in the paper to reflect changes due to updated emissions. The main effect is that SOA mass in the modeled plume is slightly lower than SOA mass in the background. This can be explained by the increase in OH and $NO_x$ which favors the fragmentation of organic compounds, reducing the amount of material available to condense to form SOA. This somewhat modifies our conclusions as now we cannot say anymore that GECKO-A is capturing an SOA increase in the plume. The comparison with the VBS parameterization becomes even more crucial to investigate which processes may be missing to explain this. With the updated emissions, the VBS still captures the observed SOA mass increase and we are still able to show the importance of adding aging to SOA processes. We can still conclude that modeling SOA in the urban plume requires including these aging processes accounted for in the VBS parameterization.

- Taking into account Comment 10, we propose a new title for the paper to better reflect its content:

    **Title:** Exploration of Oxidative Chemistry and Secondary Organic Aerosol Formation in the Amazon during the Wet Season: Explicit Modeling of the Manaus Urban Plume with GECKO-A

**Response to Anonymous Referee 1**

**Comment 1:** *Figure 1 - some readers with colorblindness may not be able to distinguish the shades of red, pink, and green used in the maps.*

**Answer 1:** We did not account for colorblindness when making this map. After checking it on an online colorblindness simulator (https://www.color-blindness.com/coblis-color-blindness-simulator/), we decided to change the color of the markers from red to black to give better contrast for colorblind people.

[Figure]

**Comment 2:** *Section 3.1.1 and Figure 2 - Line 112-113 state "The top box extends from the top of the planetary boundary layer to 1.5 km and represents the residual layer (RL). However the black dashed line in Fig 2 makes it look like the top of the second box is at $\sim 900m$.*

**Answer 2:** This was a mistake in the text. It has now been fixed to read:

**Sect. 3.1.1:** [. . . ] The top box extends from the top of the planetary boundary layer to 850 m and represents the residual layer (RL) [. . . ]

**Comment 3:** *First paragraph of 3.2.2 - the vehicle patterns and emissions in Manaus are assumed to be similar as in Sao Paolo. However, as noted in line 163-164, the fuel used in Manaus is very different than in Sao Paolo. There are many papers*

*on how ethanol blends impact vehicle emissions; the authors should at least acknowledge this literature and comment on how their assumptions about vehicle emissions might impact their results.*

**Answer 3:** We indeed assume that the VOC speciation of traffic emissions in Manaus is similar to that in São Paulo. However our emissions are scaled to the total emissions that were measured in Manaus and thus errors in emissions may come from a different distribution of vehicle types and different fuel blends between the two cities. This has an impact on the nature of emitted VOCs, which in turn would impact the SOA formation potential and oxidants lifetimes in the plume. This is now now acknowledge in the manuscript:

> **Sect. 3.2.2:** [...] The difference in the fuel blend used in São Paulo and Manaus can introduce errors in the traffic emissions VOC speciation. For instance, a recent study by Yang et al. (2019) showed that the combustion of fuels with higher ethanol content emits significantly less carbon monoxide and more acetaldehyde. Schifter et al. (2020) showed similar results, and also suggested that ethanol blends emit smaller amounts of simple aromatic compounds (*e.g.* benzene, toluene). This speciation uncertainty can especially have an impact on oxidants concentrations. Schifter et al. (2020) reported for instance that fuels containing ethanol would potentially produce less ozone after the oxidation of emitted organic species than fuels without ethanol. Moreover, the lifetime of OH is likely to change depending on the speciation of emitted VOCs due to varying reactivities with respect to OH. In the same way that the potential for ozone formation could depend on the use of ethanol fuel blends, it is also possible that the potential for SOA formation would depend on these fuel blends too. [...]

**Comment 4:** *Fig 5 - the biogenics in panel a and b show 5 hours where the box is over T3, but the anthropogenic panels c and d only show 4 hours over T3. Why is there this difference?*

**Answer 4:** As mentioned in Sect. 2, we separated T3 measurements between clean and polluted situations using the classification of T3 measurements established by de Sá et al. (2018). After hourly averaging this data, it appears that some data was positively identified as clean after 5 pm, whereas no air mass could be identified as polluted in that same time period.

**Comment 5:** *Figure 9 should be discussed more. The compositional differences between the clean and polluted cases seem very small. Is this consistent with AMS data?*

**Answer 5:** We agree this figure could be discussed more. The AMS atomic ratios data displayed on Fig. 10 also show very little difference between the clean and polluted cases. The following was added to the results discussion:

**Sect. 4.2.3:** [. . . ] The change in overall modeled SOA composition between clean and polluted cases is quite small. AMS measurements give a similar impression of a small impact of polluted situations on atomic ratios (Fig. 10), with only a slight increase of O/C ratio (see Sect. 4.2.4). Other analyses of airborne and ground AMS data (de Sá et al., 2018; Shilling et al., 2018) similarly show that the relative contribution of hydrocarbon-like organic aerosol (HOA) slightly increases in the polluted plume at the expense of isoprene derived SOA. The model and the AMS data support the idea that the impact of anthropogenic emissions is mostly seen on the total organic aerosol mass, and that all constituents of the organic aerosol phase increase approximately in the same way.

**Comment 6:** *I'm unclear on what is plotted in Fig 12b. Is this the overall species diversity from the model? Or is it the number of species needed in the reduced model to reproduce 90% of the diversity from the full GECKO model run?*

**Comment 7:** *I'm also confused about Fig 12a. Was the number of species determined by adding up the number of species needed to capture 90% of the SOA mass (i.e., simply doing amass balance)? Or was the model run in some sort of reduced form?*

**Answer 7:** We grouped these two comments as they are related.

To be clear, no reduced version of GECKO-A was developed for this work. Figure 12a depicts the number of species needed to capture 90% of the SOA mass at each timestep in the explicit model. It is made by adding up species concentrations (in decreasing order) until we reach 90% of the total modeled SOA mass.

Independantly of the calculation shown on Fig. 12a, we also calculate the statistical diversity at each timestep in the explicit model, and this is depicted on Fig. 12b.

The text in Sect. 4.4 as well as the figure caption were rephrased to explain this better:

**Sect. 4.4:** [. . . ] To answer this, two metrics are presented in Fig. 12. The first one $N_{90\%}$ is the smallest number of species needed in the explicit model to capture 90% of the total SOA mass at each timestep. After sorting species by decreasing concentration, this number is calculated by adding up these concentrations until 90% of the total modeled SOA mass is reached. The operation is repeated at each timestep. Calculated independantly, the second one is the particle diversity $D$ in the explicit modeled SOA, as defined for instance in Riemer and West (2013): [. . . ]

**Figure 12:** Smallest number of species needed to capture 90% of modeled SOA mass (left panel) with GECKO-A at each timestep ($N_{90\%}$, see text) and statistical diversity $D$ in the GECKO-A modeled particle phase (right panel, see Eq. 3).

**Response to Anonymous Referee 2**

**Comment 8:** *The focus on this work claims to be on anthropogenic influence, but one major result is the relative lack of skill GECKO-A has in capturing the impact of the Manaus plume on SOA production. This suggests to me that a major result might actually be just that the box model is poorly including anthropogenic emissions. SOA formation is underpredicted, OH concentrations are significantly over predicted, so perhaps there are just more or more reactive VOCs in the city than being included in the model.*

**Answer 8:** We agree that the disagreement between the model and measurements could result from poorly constrained anthropogenic emissions. In the course of the review process, we noticed that we were missing a significant source of anthropogenic emissions in the model, namely the emissions from the Thermal Power Plants (TPP) that provide electricity to the Manaus area and an oil refinery. Using data from Medeiros et al. (2017), we included CO, $NO_x$ and $SO_2$ additional emissions from these sources. The impact of these sources on particulate matter emissions was already intrinsically accounted for because we constrain these from the field measurements. Emissions of semi-volatile organic compounds from these sources were not included as there is not available speciation for these emissions. According to Abou Rafee et al. (2017), VOC total emissions from the power plants and the refinery are estimated to $1.6 \times 10^3$ tons $yr^{-1}$ while total VOC emissions from traffic are estimated to $2.4 \times 10^4$ tons $yr^{-1}$. We therefore can neglect them in a first approach as they are an order of magnitude lower than traffic emissions.

We updated Fig. 4a to include these emissions and added the following description in the text:

> **Sect. 3.2.2:** [. . .] Additionally, emissions from 11 local thermal power plants (TPP) and one oil refinery located in the vicinity of Manaus were obtained from the data presented in Medeiros et al. (2017). Based on monthly statistics of fuel use in each of the TPP and the oil refinery, combined with emission factors of CO and $NO_x$ for each type of fuel (diesel, fuel oil, natural gas), we calculated CO and $NO_x$ emissions for February, March and April 2014. These total emissions were then averaged over the whole surface area of Manaus (377 $km^2$, Abou Rafee et al., 2017). Total SO2 emissions were taken from Abou Rafee et al. (2017) and added to the urban emissions for the considered Manaus area.

We ran the simulations with these new emissions. Their main impact is to increase OH and NOx concentrations in the plume. The impact of urban emissions on SOA mass is weaker with the new emissions. This is probably due to the combination of (i) higher NOx mixing ratios, which reduce biogenic SOA through enhanced fragmentation of alkoxy radicals in the gas phase, and (ii) the contribution of anthropogenic compounds. We have updated the corresponding figures and numbers to account for the new simulations, however our conclusions remain unchainged.

This shows the sensitivity of the model to urban emissions, but after our fix our emissions are quite similar to those used in other modeling studies (Abou Rafee et al., 2017; Shrivastava et al., 2019).

**Comment 9:** *This is further exacerbated by the issue that the surrogate composition of the fuel emissions are known to not really accurately represent the true composition. The ability of the model to capture clean conditions, which with MEGAN and*

*the PTR have reasonably well constrained emissions, speaks well of GECKO-A, and suggests just that the issue may be in the emissions being fed to the model.*

**Answer 9:** It is also true that the fuel composition is not accurately known, especially for IVOCs. We address this issue in
our answer to Comment 3 from the first anonymous referee.

**Comment 10:** *This paper is primarily an exploration of the strengths and weakness of GECKO-A,it doesn't do much to discuss the "Impact of Urban Emissions" as the title suggests. There is some discussion, but mostly it is a comparison of GECKO-A to other models (VBS, Shrivastava) and not really providing new information beyond a further exploration of GECKO-A. I think there are interesting results regarding GECKO-A, in particular the discussion of reduced complexity, which seems like a real place for GECKO-A to provide generalizable scientific insight (how complex do we really need models to be?), and that the weaker part of the work is the attempts to understand the Manaus plume (for which the emissions may not be correct, and aging may not be incorporated,and other issues). Potential issues in modeling the plume end up conflated with potential issues in the model, and instead I think perhaps some of the discussions about the model could instead be bolstered and thought about more deeply (e.g., I don't think the explanations for H/C discrepancies are likely complete).*

**Answer 10:** After noticing the inability of GECKO-A to reproduce the observed urban SOA enhancement, we indeed focused more about finding reasons for this behavior than exploring the modeled impact of Manaus emissions, hence the comparison with the simpler model from Shrivastava et al. (2019). Nevertheless this investigation still gives some reasons to believe that the in-particle aging of organic aerosol is an important part of the interaction of urban emissions with clean biogenic air masses. In our answer to Comment 8, we highlight how we addressed a possible problem with urban emissions. We don't have yet the tools in GECKO-A to provide more information on the specific processes involved in aging of particles, but following this comment we extended the discussion of the H/C discrepancies with additional calculations to estimate the possible impact of dimerization and fragmentation:

**Sect. 4.2.4:** [...] As a test, we generalized this estimation to all $C_{10}$ in the aerosol phase: we replaced each $C_{10}$ by the corresponding $C_{20}$ and halved its concentration. In this way, we can calculate what would H/C and O/C ratios be in the aerosol phase if aging processes only dimerized $C_{10}$ compounds. The resulting modeled van Krevelen diagram is reported on Fig. 10 (labeled w/ dimer.). The impact of $C_{10}$ dimerization is relatively strong on O/C ratio, ranging from 0.66 to 0.78 and remaining in the range of measured O/C ratios at T3 site and in the aircraft. H/C ratios are only reduced to 1.88–1.94, still 50% higher than measured H/C at the T3 site and 20% higher than airborne data.

[...]

As another test, we also estimated what would O/C and H/C ratios be if all $C_{10}$ fragmented in the aerosol phase. The resulting modeled van Krevelen diagram is reported on Fig. 10 (labeled w/ frag.). In this case, modeled O/C ratios increase to a range of 0.88 to 0.96 and remain in the higher end of measured ratio at the T3 site. H/C are reduced further than in the dimerization test and sit at the higher end of airborne measured H/C ratios, but they still are 45% higher than H/C ratios measured at the T3 site.

Even if they apparently cannot account for the discrepancy between modeled and measured H/C ratios, the two tests presented here on $C_{10}$ compounds in the aerosol phase show the potential importance of adding these missing processes in GECKO-A. These simple tests are however simplifications that overlook important factors in the potential impact on SOA composition: (i) not all $C_{10}$ compounds would be affected by these processes, (ii) other compounds than $C_{10}$ could react in a similar way, (iii) trimerization, tetramerization and other accretion processes could also occur in the aerosol phase, (iv) missing fragmentation processes could also happen in the gas phase.

**Comment 11:** *Line 1. Missing "of" : "investigation of the"*
**Answer 11:** Fixed

**Comment 12:** *Line 7. Not clear what it means for the model to "reproduce measured primary compounds" after tuning emissions. Aren't the primary compounds just the emissions, so you tune for this result, and it isn't really impacted by the skill of the model? Perhaps it will be more clear to me after a detailed reading.*

**Answer 12:** We agree this mention of primary compounds is of little interest to the abstract. The sentence was modified as follows:

**Abstract:** [...] The biogenic emissions estimated from existing emission inventories had to be reduced to match measurements. The model is able to reproduce ozone and $NO_x$ for clean and polluted situations. [...]

**Comment 13:** *Line 15. "particularly intense all year long" is a bit odd, perhaps the authors mean "more photochemically active than other regions throughout most of the year"?*

**Answer 13:** We agree and used this suggestion in the text.

**Comment 14:** *Line 36-37. This mid-paragraph question is odd. "Would" in what case? Do the authors mean "Do"? Maybe just rephrase this without the use of this rhetorical question.*

**Answer 14:** We agree and rephrased the sentence as follows:

> **Introduction:** [. . . ] Several studies have investigated how the biogenic nature of the SOA is affected by anthropogenic influence. [. . . ]

**Comment 15:** *Line 50. The discussion of the "molecular view," while valuable, has some gaps or issues here. In particular, those citations of Koss et al. are an odd choice, as I believe they are just using a PTR, and if I'm not mistaken, there was a PTR run at T3 by the Martin group (some of the data of which is used in this work). Other groups (e.g. the Kroll group, and the CLOUD group) have tried to combine multiple spectrometers to actually capture the whole range of compounds, reaching more a molecular view. However, even still, this would be more reasonably considered a "formula view", as these instruments do not separate molecules out from their formulas. There were also several other instruments at T3 approaching a molecular view (I believe for instance the Goldstein group has collected and run GCxGC of filter samples, providing some molecular invormation), though perhaps not comprehensively. I would re-frame some of this discussion to more accurately capture the landscape and discuss what existing measurements can or can't provide (e.g., I agree the available instruments probably don't provide a comprehensive measurement to compare to models, particularly for gas-phase oxygenates).*

**Answer 15:** We agree that we should make the distinction between the "molecular" and "formula" views. We in fact mixed these and called them both "molecular" views. We reworked this part of the introduction to account for this comment, and introduced the term "formula" view:

> **Introduction:** [. . . ] In a recent review, Heald and Kroll (2020) have reported on the recent progress in measurements of individual organic compounds, and how experimentalists are getting close to achieving closure on organic carbon in both gas and aerosol phases (*e.g.* Gentner et al., 2012; Isaacman-Vanwertz et al., 2018). As these measurements are able to capture elemental formulas, double bonds, some oxygenated functional groups and aromaticity (*e.g.* Yuan et al., 2017), they do not provide individual molecular identities. From this point of view, measurements are still restricted to a "formula view". For the GoAmazon field campaign, Yee et al. (2018) were able to sample and identify 30 sesquiterpenes and 40 of their oxidation products at the T3 site with a semi-volatile thermal desorption aerosol gas chromatograph (SV-TAG, Isaacman et al., 2014) but they do not achieve the coverage needed to approach the "molecular view". [. . . ]

**Comment 16:** *Line 54. I'm a big fan of GECKO-A, but maybe not everyone will agree that it is "the ideal tool". "an excellent tool" perhaps?*

**Answer 16:** We are big fans of GECKO-A too, and we may have got carried away with our enthusiasm. The text now reads:

**Introduction:** The Generator for Explicit Chemistry and Kinetics of Organics in the Atmosphere (GECKO-A, Aumont et al., 2005; Camredon et al., 2007) is an excellent tool to model atmospheric organic chemistry with a detailed molecular view.

**Comment 17:** *Line 171. The authors discuss the fact that n-alkanes are perhpas not a good surrogate for diesel fuel and gasoline IVOCs, which are mostly branched and cyclic. However,they do not explain why their estimates are less branched than other work has suggested, and importantly they do not disucss the impacts of these structural differences. They acknowledge that almost none of diesel fuel is comprised of the compounds they are using as surrogates, but do not further discuss this issue. Gentner et al. attempt to put estimates on the impact of branching and rings on SOA, so estimating this uncertainty shouldn't be too difficult.*

**Answer 17:** This IVOC alkane surrogate speciation was first established in Lee-Taylor et al. (2011) to obtain a good volatility distribution, because the identity of individually emitted species was not well known. We kept this simplified IVOCs emissions mostly because of mechanism size issues. Adding heavier molecular weight branched alkanes, cycloalkanes, linear and branched alkenes and aromatics matching the distribution displayed in Gentner et al. (2017) would increase the size of the mechanism by a factor of 5 or 6, to an unmanageable size even by GECKO-A standards. We agree that the possible impact of this simplification should be discussed and we added the following to the description of Manaus emissions:

**Sect. 3.2.2:** [...] Choosing alkanes as surrogates for emitted IVOCs is also likely to introduce uncertainties to SOA produced from their oxidation. Lim and Ziemann (2009) carried out multiple chamber experiments that investigated the impact of branching and rings on alkanes SOA yields. For instance they showed that SOA yields range from a few percent for branched alkanes with 12 carbon atoms to $80\%$ for cyclododecane while n-dodecane has an SOA yield of $\approx 32\%$. La et al. (2016) simulated these experiments with GECKO-A and they were able to reproduce this experimentally observed behavior. This means that without a detailed inventory of emitted IVOCs, the uncertainty on the SOA yield from IVOCs is high in our version of the model. It should be noted that the range of measured SOA yields for structurally different compounds with the same number of carbon atoms seems to peak for $C_{10}$-$C_{13}$ alkanes. The range of observed SOA yields in Lim and Ziemann (2009) decreases after this peak. For instance, SOA yields for $C_{15}$ alkanes of various structures range from $45\%$ to $90\%$. We can therefore expect the IVOCs SOA yield to be highly sensitive to the speciation of compounds ranging from $C_{12}$ to $C_{14}$, but this sensitivity should decrease for heavier molecular weight species. [...]

**Comment 18:** *Figure 5. The measurement of benzene and toluene at T3 are fairly poor constraints on these species as they catch only the tail end of the decay (and even still don't really agree with the toluene model). Is there no aircraft data or VOC measurements at T1 or T2 near Manaus to better constrain these?*

**Answer 18:** There were measurements of benzene and toluene taken in the aircraft, but to our knowledge no data is available for these compounds in the sites in or close to Manaus. Airborne measurements were carried out in the plume and we added the corresponding points on the Fig. 5. The figure caption was updated:

**Figure 5:** Modeled (lines, second day) time evolution of primary species concentrations in the Lagrangian box-model described in Sect. 3.1, average experimental concentrations measured at the T3 site (dots) and in the airplane (triangles). The vertical range of the experimental data denotes the standard deviation of measured concentrations during events identified as clean (top, blue) and polluted (bottom, orange). The airborne data was measured during plume transects. For each transect, aircraft distance from Manaus was converted to a time separation from Manaus assuming the plume leaves the city at 8am and arrives above T3 at 2pm.

It should be noted that modeled benzene and toluene lines were switched on the original Fig. 5. We fixed that too and updated the text to account for this:

**Sect. 4.1:** The modeled mixing ratio of benzene matches the measurements, between 0.4 and 0.6 ppb, while modeled toluene is closer to the higher range of measurements, between 0.2 and 0.6 ppb during the afternoon. Figure 5 also displays the airborne measurements of the same anthropogenic compounds. The modeled mixing ratios of benzene and toluene decay in a similar way to the concentrations measured during the plume transects. The modeled peak is not seen by the aircraft measurements as the aircraft may not be flying close enough to the emission sources to capture it.

**Comment 19:** *Line 260. Again, are there T1 or T2 measurements that could help constrain VOCs in the city? Also, to what extent does model include residual biogenic VOCs present in the city? Presumably there are VOCs present in the city other than just the vehicle emissions, like biogenics from the surrounding forest (or volatile chemical products?), are those captured by the model? Is there a spin up time to allow the city emissions to have some equilibrium concentration of VOCs that would help suppress OH concentrations?*

**Answer 19:** We are not aware of measurements at T1 or T2 that could have helped constrain VOCs in the city itself. However the comparison to airborne measurements mentionned in Comment 18, provides an additional constraint. This two boxes boxmodel used in this study is designed to simulate an air mass traveling over the rainforest. The bottom box is then exposed to fresh Manaus emissions for 1 h, the approximate time it would take for an air mass traveling at 10 m s$^{-1}$ to cross the urban area. This model design does not require the city chemistry to be at equilibrium before interaction with the box.

After the emission update presented in Comment 8, the two major sources of anthropogenic pollution are accounted for with traffic and power plants emissions. To our knowledge the potential contribution of personal care products VOCs emissinos to anthropogenic emissions has only been evaluated in North America (*e.g.* Coggon et al., 2018; McDonald et al., 2018; Shah et al., 2020). This contribution is likely to become relatively important in the future with the decrease of vehicle emissions in western developed countries, but is not likely to be important in Manaus in 2014 and 2015.

The interaction of biogenics from the surrounding forest with urban emissions is exactly what happens in the model as soon as it is exposed to Manaus emissions: biogenic emissions are replaced with urban emissions for a short time, but the urban emissions become mixed with the remaining background from biogenic chemistry.

**Comment 20:** *Figure 8. It would be helpful to add Glasius OS measurmenets to the figure as a dashed line.*

**Answer 20:** D one.

**Comment 21:** *Line 300-314. I'm not sure the explanations provided for the H/C disparity can really close the gap and it warrants further discussion. The examples the authors provide of oligomerization and fragmentation provide relatively modest decreases in H/C and also change the O/C. A huge fraction of the compounds would need to be oligomers or fragments for this to reconcile H/C, and this would likely shift the O/C. In the examples they provide, dimers still have H/C ratios well above the observed average. If, for instance, all of the C10 compounds were actually dimers, wouldn't this just bring the average down to 1.5 or 1.6? How do you get down to 1.3? What sorts of compounds can push H/C this far down? Trimers? Tetramers? And would the whole mass needto be comprised of these? The authors could play some games with their data to explore this, for instance assume all compounds are actually dimers or fragments and estimate the average H/C and O/C. This might further be scientifically interesting by putting some constraints on accretion products.*

**Answer 21:** As presented in Comment 10, we introduced the calculation of what would O/C and H/C ratios become after dimerization or fragmentation of $C_{10}$ compounds in the aerosol phase. These processes seem to not be sufficient to bring modeled H/C ratios down to measured values, especially if we don't want to move modeled O/C ratios too far from measured values too.

**Comment 22:** *Line 316. How do the anthropogenic emissions used by Shrivastava compare to those used here?*

**Answer 22:** Shrivastava et al. (2019) followed an approach very similar to this work to estimate urban emissions. Their work is also based on combining data from Manaus and São Paulo for traffic and power plants emissions. We completed the description of their simulation to clarify this:

> **Sect. 4.3:** Shrivastava et al. (2019) modeled this same field campaign with WRF-Chem, a chemistry transport regional model (Grell et al., 2005) and similarly to this work they based their primary organic compounds emissions on the MEGAN inventory (Guenther et al., 2012) for biogenic compounds, and combining the methodology described in Andrade et al. (2015) with data from Medeiros et al. (2017) for anthropogenic emissions. [...]

**Comment 23:** *Line 322. I don't fully understand the aging parameterization, could the authors provide more detail?*

**Answer 23:** We added the following clarification to the text:

> **Sect. 4.3:** [...] This aging is parameterized as a reaction of each of the SOA surrogate with OH as follows:
>
> $$VBS_n + OH \rightarrow \alpha_{frag}VBS_{n+1} + (1 - \alpha_{frag})VBS_{n-1} \qquad (R1)$$
>
> The reaction rate is $k_{R1} = 2 \times 10^{-11}$ cm$^3$ molec$^{-1}$ s$^{-1}$. The branching ratio for fragmentation $\alpha_{frag}$ is determined as the ratio of the reaction rate of peroxy radicals with NO to the sum of all peroxy radical reactions rates; it has an upper limit of 75%. [...]

**Comment 24:** *Line 362. The authors point out the importance of aging in capturing the polluted SOA,which the GECKO-A model does not really capture. Is this due to a lack of aging in the GECKO-A model? Maybe I missed it, but is the GECKO-A model only oxidizing the gas phase and not aging the particle? Considering the importance of aging on reproducing the SOA mass (Figure 7), could the authors include a parameterization of aging in the GECKO model? This might also address the H/C and O/C issues.*

**Answer 24:** There is no aging parameterization in GECKO-A, oxidation only happens in the gas phase. For technical reasons and lack of resources, we were not able to implement a parameterization of aging in GECKO-A for this paper. In our answer to Comment 10, we estimate the possible impact of dimerization and fragmentation of biogenic condensed species on H/C and O/C but we don't see how to easily estimate the potential impact on SOA mass.

**Comment 25:** *Line 367-368. Why is it unclear? It seems to capture polluted periods better than biogenic periods (Figure 7).*

**Answer 25:** The VBS parameterization is capturing both polluted and biogenic episodes relatively well, but in both situations SOA composition is dominated by biogenic oxidation products. In that sense, we don't know if the same parameterization would apply well in environments where the composition of SOA would be for instance dominated by species of anthropogenic or biomass burning origin.

**Comment 26:** *Line 400-402. The ideas of reducing complexity discussed in this section are very interesting. It's not completely clear to me that some of the conclusions aren't overextended.In particular, the conclusion that "diversity represents the number of species that would be needed to reproduce the same informational content regarding the composition of SOA." The parameter D is based solely on mass fraction, not their physicochemical properties, so it might not capture other properties such as oxygenation or volatility.Imagine a scenario where a small fraction of highly oxygenated compounds drive up O/C - this might still impact hygroscopicity but would this complexity be captured by D? Or a small fraction of more volatile components might partition between the gas-and particle-phase and drive oxidation of particle mass. While the exact mechanism of implementing such a reduction is out of scope of this mansucript, the overall point is that it's not to me that either of the diversity parameters really capture the complexity. Would it be possible to implement a variant of D for parameters of interest? For instance minimum number of components needed to describe O/C within 10% while capture some fraction of the mass? I'm not sure the best parameters, but the idea would be something that captures some of the properties beyond simply mass.*

**Answer 26:** This is a very interesting take on the reduction issues. We agree with the idea that this diversity approach should be extended to capture other properties than just mass and that we overextended the conclusion about the composition of SOA. We fixed the sentence you mention as follows:

> **Sect. 4.4:** [. . . ] As this number is directly derived from informational entropy, we suggest that the diversity represents the number of species that would be needed to reproduce the same informational content regarding the time evolution of SOA mass in the explicit model. [. . . ]

For this paper, we could not explore further this application of information theory to explicit mechanisms reduction. Hopefully this will be the subject of future work. We added the following to the discussion in that same section, raising the issues mentioned in this comment.

> **Sect. 4.4:** [. . . ] Finally, we used in this section an entropy calculation for SOA mass: it is based only on mass fractions of the species composing the modeled organic particles. The effective number of species displayed on Fig. 12 is therefore only meaningful for SOA mass and properties directly linked to it. If the goal is to predict other properties, *e.g.* hygroscopicity, toxicity or optical properties, assuming we find a way to calculate these with GECKO-A, the diversity defined here would not necessarily be meaningful. For instance, hygroscopicity or toxicity could be driven by a handful of oxygenated species that do not matter for the informational content regarding SOA mass. We did not explore further down this path, as this is not the main subject of this paper, but it may be possible to generalize this definition of informational diversity to properties other than mass.

As an additional note, the idea is not to reduce complexity as written at the beginning of this comment. It is more about evaluating complexity, and showing how many species are needed to effectively produce the same complexity (regarding SOA mass in our example) as the explicit mechanism.

**Comment 27:** *Line 403-405. Isn't it not only possible but certain that an effective species is not an individual species but rather a combiniation of explicit species? Otherwise D would be more similar to the other metric, or equal to N. The very nature of the parameter D is to be a mathematical descriptor, not an individual species, correct?*

**Answer 27:** Thanks for this comment. Intuitively your reasoning makes sense and we added our understanding of it to that section (additionally, $N_{90\%}$ has been introduced to denote the "90%" metric):

> **Sect. 4.4:** [. . . ] For instance, in the polluted scenario, $D$ is a factor of 7 lower than $N_{90\%}$. This should mean that $D$ cannot represent a subset of the individual species from the original mechanism, otherwise it would be expected to be equal or higher than $N_{90\%}$ if it is supposed to reproduce the informational content regarding SOA mass. It is therefore likely, and making this problem more complex, that each of these effective species is a (non) linear combination of explicit individual species. [. . . ]

However we are trying to remain very cautious about what we write on the interpretation of this metric as we are not experts in the field of information theory.

[revised manuscript text omitted]

---

## Author Response (AR2)

**Response to Minor Comments from Anonymous Referee #2 Impact of Urban Emissions on a Biogenic Environment during the wet season: Explicit Modeling of the Manaus Plume Organic Chemistry with GECKO-A**

Camille Mouchel-Vallon et al.

We thank the reviewer for their helpful comments. Below is our detailed answer to their specific comments, followed by the marked up manuscript.

**Response to Minor Comments**

Comment 1: Page 7, lines 150-159 - (a) Please clarify the tuning of the MEGAN emissions. Isoprene emissions were adjusted by a factor of 7 - presumably this was to make the modeled concentrations match the PTRMS measurements. (b) How were monoterpenes assigned to the different isomers? The PTRMS data is not a helpful guide in this case, because it cannot differentiate e.g., limonene from a-pinene.

Answer 1: Isoprene emissions were indeed adjusted to get the modeled concentrations to match the PTRMS measurements on the ground at T3. Total monoterpenes emission was also adjusted to match the PTRMS measurements at T3. As we mention

10 in the text, we speciated the monoterpenes based on data collected by Jardine et al. (2015), who used TD-GC-MS measurements to measure vertical profiles of 12 different monoterpenes on a mast in a remote rainforest location. We modified the text to clarify this.

**Sect. 3.2.1:** [...] Monoterpenes were then speciated to match concentrations measured by Jardine et al. (2015) at the top of an Amazonian rainforest canopy with a thermal desorption-gas chromatograph-mass spectrometer (TD-GC-MS). Based on this emission inventory, we then simultaneously optimized isoprene and total monoterpenes emissions to match the model with isoprene and total monoterpenes measured at T3 under clean conditions. [...]

**Comment 2: Line 285 - I am not sure what "see 2" means.**

Answer 2: We meant to refer to Sect. 2, where we explains how clean and polluted experimental data where identified. This has been fixed in the text.

**Comment 3:** Figure 7 and section 4.2.1: Line 300 states that measured SOA in the clean case ranges from 0.6 to 2.5  $ug/m^3$ . These seem to be the extremes, not the averages of the clean periods. The authors argue that the model performs better during

the clean period than the polluted period. However, it looks like the blue and orange model lines run about halfway between the clean and polluted data (with the exception that the model seems to do better for the 12pm clean point). Therefore it is not

20 *obvious that the model performs better for clean conditions than polluted.*

Answer 3: L ooking at Fig. 7, to us it looks like in the clean case between 12 pm and 4 pm, the model at least falls in the range of the mean values  $\pm 1$  std. dev. (the vertical range on the plot depict the standard deviation of the experimental data). This is never the case for the polluted case simulation. So even if the clean model does not match the mean experimental value, it is still within one standard deviation of the measured data which is better than the polluted model.

**25 **Comment 4:** *Line 310-311 first note that organosulfates are over predicted relative to the Glausius measurements, but that the over prediction is consistent with the same paper. Please clarify.**

**Answer 4:** S imilarly to the previous comment, even if the model does not match the mean value measured in Glasius et al. (2018), it is still between the mean and the mean + 1 std. dev. (depicted by the vertical range in Fig. 9). This is what we refer to when we mention that the model values are "in the higher range of the reported measured values".

**30 Comment 5:** *Figure 9 - what does the ">CO" functional group stand for?**

**Answer 5:** The >CO notation designates ketone functional groups, to be differentiated from aldehyde groups (-CHO). We made the definition of the functional groups explicit in the figure caption.

**Fig. 9:** [...] Functional groups are abbreviated as follows: aldehyde (-CHO), carboxylic acid (-CO(OH)), hydroxy (-OH), nitrate (-ONO2), hydroperoxide (-OOH), sulfate (-OSO3) and ketone (>CO). [...]

[revised manuscript text omitted]